# Characterization of a KDM5 small molecule inhibitor with antiviral activity against hepatitis B virus

Sarah A. Gilmore[1¤a], Danny Tam[1], Tara L. Cheung[1], Chelsea Snyder[1], Julie Farand[1], Ryan Dick[1¤b], Mike Matles[1], Joy Y. Feng[1], Ricardo Ramirez[1], Li Li[1], Helen Yu[1], Yili Xu[1], Dwight Barnes[1], Gregg Czerwieniec[1¤c], Katherine M. Brendza[1¤c], Todd C. Appleby[1], Gabriel Birkus[1¤d], Madeleine Willkom[1], Tetsuya Kobayashi[1], Eric Paoli[1¤e], Marc Labelle[2†], Thomas Boesen[2,3], Chin H. Tay[1¤f], William E. Delaney, IV[1¤g], Gregory T. Notte[1], Uli Schmitz[1]*, Becket Feierbach[1]

**1** Gilead Sciences, Inc., Foster City, California, United States America, **2** EpiTherapeutics ApS, Copenhagen, Denmark, **3** Novo Nordisk A/S, Bagsvaerd, Denmark

† Deceased. M.L. passed away before the submission of the final version of this manuscript. U.S. accepts responsibility for the integrity and validity of the data collected and analyzed.
¤a Current address: Allovir, Waltham, Massachusetts, United States America
¤b Current address: Maze Therapeutics, South San Francisco, California, United States America
¤c Current address: Nektar Therapeutics, San Francisco, California, United States America
¤d Current address: Institute of Organic Chemistry and Biochemistry of the Czech Academy of Sciences, Prague, Czech Republic
¤e Current address: Verily Life Sciences, South San Francisco, California, United States America
¤f Current address: Vir Biotechnology, San Francisco, California, United States America
¤g Current address: Assembly Biosciences, South San Francisco, California, United States America
* uli.schmitz@gilead.com

**Data Availability Statement:** All relevant data are within the paper and its Supporting information files.

## Abstract

Chronic hepatitis B (CHB) is a global health care challenge and a major cause of liver disease. To find new therapeutic avenues with a potential to functionally cure chronic Hepatitis B virus (HBV) infection, we performed a focused screen of epigenetic modifiers to identify potential inhibitors of replication or gene expression. From this work we identified isonicotinic acid inhibitors of the histone lysine demethylase 5 (KDM5) with potent anti-HBV activity. To enhance the cellular permeability and liver accumulation of the most potent KDM5 inhibitor identified (GS-080) an ester prodrug was developed (GS-5801) that resulted in improved bioavailability and liver exposure as well as an increased H3K4me3:H3 ratio on chromatin. GS-5801 treatment of HBV-infected primary human hepatocytes reduced the levels of HBV RNA, DNA and antigen. Evaluation of GS-5801 antiviral activity in a humanized mouse model of HBV infection, however, did not result in antiviral efficacy, despite achieving pharmacodynamic levels of H3K4me3:H3 predicted to be efficacious from the in vitro model. Here we discuss potential reasons for the disconnect between in vitro and in vivo efficacy, which highlight the translational difficulties of epigenetic targets for viral diseases.

**Funding:** The author(s) received no specific funding for this work.

**Competing interests:** The authors have declared that no competing interests exist.

## Introduction

Chronic hepatitis B (CHB) is a major global health care challenge and one of the main causes of liver diseases including cirrhosis and hepatocellular carcinoma (HCC). Of the estimated 2 billion people acutely infected with Hepatitis B virus (HBV), approximately 240 million people develop CHB and 880,000 people die annually of complications from CHB [1–3]. Nucleos(t)ide analogues and interferon-α (IFN-α) are approved treatments for CHB and result in suppression of viral replication; however, current treatment regimens rarely result in a functional cure [4]. Thus, novel antiviral therapies that can cure CHB patients are needed.

HBV is a small 3.2 kb DNA virus that infects hepatocytes in the human liver. Upon infection of hepatocytes, the HBV genome enters the nucleus and is converted into covalently closed circular DNA (cccDNA). cccDNA is a stable, chromatinized episome that serves as the template from which all viral RNA is transcribed [5, 6]. Nucleos(t)ide and IFN-α therapies do not directly target cccDNA and existing long-term therapy fails to significantly impact cccDNA reservoirs in the majority of patients [4, 7]. Therefore, strategies that eliminate cccDNA or effectively silence the transcription of viral antigens are needed.

Emerging evidence suggests that the transcription of cccDNA is governed by the accessibility of its chromatin structure. Further, posttranslational modification of histones dynamically regulates cccDNA structure [8–11], akin to the well-studied epigenetic regulation of eukaryotic genomes [12]. Agents that interfere with the epigenetic control of cccDNA may disrupt transcription of viral genes and subsequently prevent the production of viral antigens leading to a functional cure of CHB. To this end, we embarked on a screen in a primary human hepatocyte (PHH) model of HBV infection for compounds that repressed cccDNA transcription using a focused library of known epigenetic modulators [13]. From this screen we identified two classes of small molecules with antiviral activity against HBV, retinoids (ref) and inhibitors of lysine demethylase 5 (KDM5).

The KDM5 family (KDM5A –D or JARID1A –D) is a member of the Jumonji C (JmjC) domain containing demethylases, which catalyze the demethylation of histones in an iron (II) and α-ketoglutarate dependent manner [14, 15]. KDM5 specifically demethylates the mono-, di-, and trimethylated lysine 4 residue of histone 3 in nucleosomes (H3K4me, H3K4me2, and H3K4me3) (see Fig 1A) [16, 17]. In eukaryotic genomes, H3K4me3 is predominantly localized at the transcription start sites (TSS) of highly-expressed genes, where it plays a role in RNA polymerase II binding and target gene activation [18–22]. KDM5 family members are predicted to act as transcriptional repressors based on the presence of H3K4me3 at the promoters of most actively transcribed genes [23]. However, KDM5 may serve a broader function to promote appropriate transcription by demethylating H3K4 in gene bodies thereby focusing H3K4 methylation at TSS [23, 24]. Accordingly, KDM5 has become an important oncology target as is evident from several dozen of inhibitor patents and other publications [25, 26].

In this work we describe the discovery that a potent small molecule inhibitor of KDM5, termed GS-080 [27–30], has antiviral activity against HBV in a PHH infection model. An ethyl ester prodrug of GS-080 (Fig 1), termed GS-5801, was utilized to increase the cellular permeability, oral bioavailability, and liver-loading of the parent GS-080 molecule. Treatment of HBV-infected PHH with GS-5801 causes accumulation of H3K4me3 relative to total H3 (H3K4me3:H3 ratio) on cellular DNA that correlates with a reduction in HBV RNA, DNA, and antigens. In vivo studies in rats and cynomolgus monkeys show that GS-5801 is well-tolerated and selectively promotes the increase of the H3K4me3:H3 ratio in the liver to a greater extent than other tissues. Evaluation of GS-5801 antiviral activity in a humanized mouse

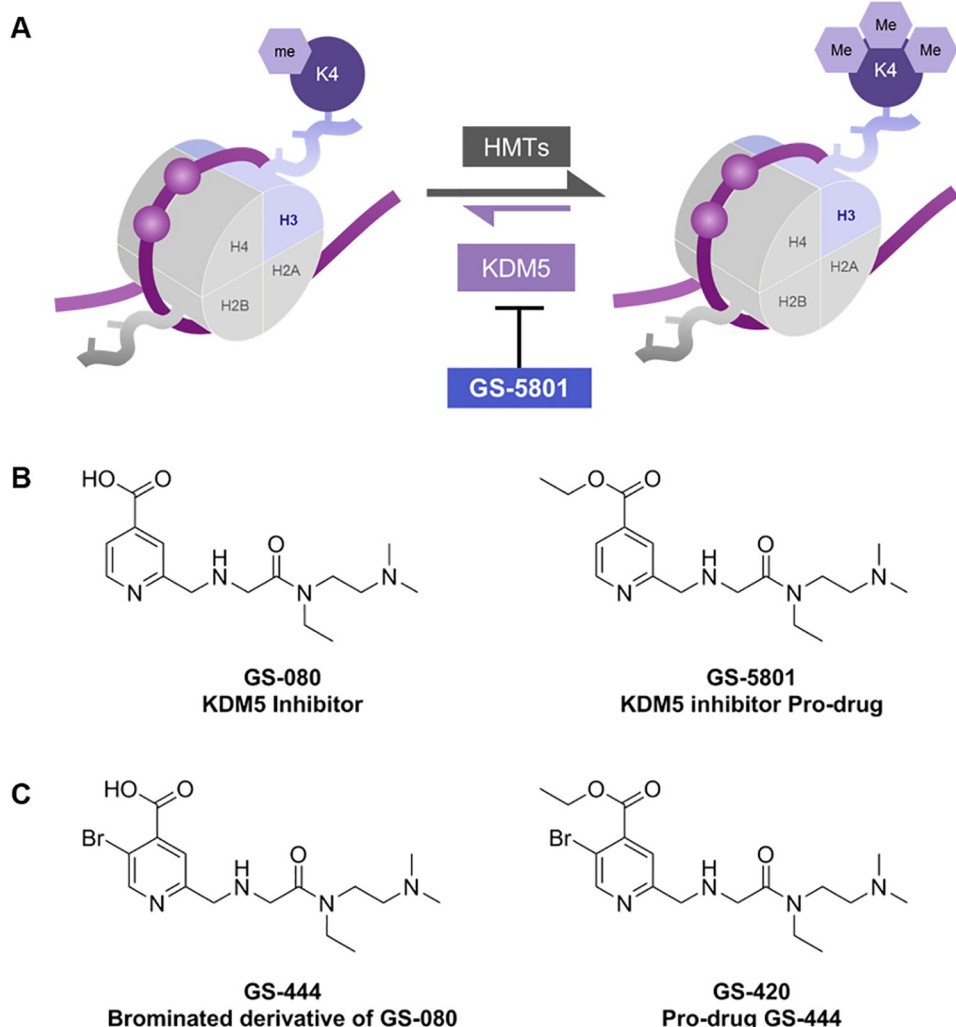

**Fig 1. GS-5801 is a prodrug of KDM5 inhibitor GS-080.** (A) KDM5 demethylates the lysine 4 of the histone 3 (H3K4) subunit of nucleosomes. Inhibition of KDM5 activity by compounds such as GS-5801 results in accumulation of methylated H3K4 on chromatin through the activity of cellular histone methyltransferases (HMT) that catalyze the mono-, di-, and tri- methylation of H3K4. Chemical structures of prodrug GS-5801 and parent GS-080 (B) as well as prodrug GS-420 and parent GS-444 (C) are shown.

model of HBV-infection, however, did not result in efficacy at doses predicted to be efficacious based on GS-5801 in vitro antiviral activity and pharmacodynamics (H3K4me3:H3). Together these data highlight discordance between the antiviral effects of GS-5801 observed in HBV-infected primary human hepatocytes and a humanized mouse model of HBV infection. Understanding the translatability of therapeutic agents in HBV infection models to chronic hepatitis B infection is valuable, especially given the limited number of host targets that have been evaluated clinically in CHB patients. Despite a high risk of the in vivo study predicting lack of clinical efficacy, we pursued the evaluation ofGS-5801 in Phase 1a and Phase 1b clinical trials for chronic hepatitis B. As GS-5801 is still the first and only KDM5 inhibitor to have reached the clinic, our experience underscores the translational challenges with epigenetic targets.

## Results

### GS-5801 inhibits HBV RNA, DNA, and antigens in primary human hepatocytes

To identify small molecule compounds capable of inhibiting HBV transcription, a targeted library of epigenetic modifiers was evaluated in HBV-infected PHH to identify compounds that reduced HBV intracellular RNA and secreted antigens: hepatitis B virus e antigen (HBeAg) and hepatitis B virus s antigen (HBsAg). From this screen we identified nicotinic acid derivative GS-080, and its more cell-permeable prodrug GS-5801 (see Fig 1B) [27–30], as inhibitors of HBV RNA and antigen production in HBV-infected PHH. Importantly, neither GS-5801 nor GS-080 had measurable cytotoxicity in PHH or in a panel of human cells at concentrations up to 57 μM (see **S1 Table in** S1 File).

To further characterize the antiviral activity of GS-5801, PHH were infected with HBV (genotype D; GTD) for three days prior to initiation of GS-5801 dosing. HBV-infected PHH received a dose of GS-5801 every three to four days for a total of four doses over 14 days. HBV intracellular RNA, extracellular DNA, and secreted HBsAg and HBeAg were measured on Day 14 (time post-initiation of dosing). GS-5801 $EC_{50}$ values ranged between 0.034–1.1 μM (median $EC_{50}$ = 0.16 μM) for intracellular RNA, 0.0071–1.3 μM (median $EC_{50}$ = 0.14 μM) for extracellular DNA, 0.015–1.7 μM (median $EC_{50}$ = 0.24 μM) for secreted HBsAg, and 0.014–1.1 μM (median $EC_{50}$ = 0.15 μM) for secreted HBeAg in PHH with no cytotoxicity ($CC_{50}$ > 10 μM) in the seven PHH donors tested (Table 1). As a negative control, GS-444, a bromo-derivative of GS-080 with vastly reduced KDM5 inhibitory activity (GS-444 $IC_{50}$ for KDM5B = 2760 nM; 7000-fold reduced over GS-080), was administered to PHH as its cell-permeable pro-drug GS-420 (Fig 1C). GS-420 did not exhibit HBV antiviral activity in PHH at concentrations up to 10 μM (**S2 Table in** S1 File).

Amounts of cccDNA, the DNA template for transcription of HBV antigens and pre-genomic RNA, were measured to determine whether the reduction of intracellular HBV RNA,

**Table 1. Biochemical potency and selectivity of GS-080 against KDM enzymes.**

| KDM Enzyme | Enzyme Concentration (nM) | Positive Controls $IC_{50}$ (nM)[a] | $IC_{50}$ (nM)[b] | Fold Selectivity Compared to KDM5A[c] |
|---|---|---|---|---|
| KDM5A | 2.5 | PDCA (410) | 0.36 | 1 |
| KDM5B | 1.2 | PDCA (410) | 0.38 | 1 |
| KDM5C | 1 | IOX1 (990) | 3.7 | 10 |
| KDM5D | 5.5 | PCDA (75) | 66 | 183 |
| KDM4A | 0.2 | PDCA (870) | 7.2 | 20 |
| KDM4B | 1 | PDCA (860) | 4.7 | 13 |
| KDM4C | 1 | PDCA (510) | 4.8 | 13 |
| KDM1A | 0.25 | S2101 (2000) | > 100,000 | > 278,000 |
| KDM2B | 2 | PDCA (19000) | 620 | > 1,720 |
| KDM3A | 0.3 | IOX1 (140) | 1,700 | > 4,720 |
| KDM3B | 0.1 | PDCA (7900) | > 10,000 | > 27,800 |
| KDM6A | 2 | PDCA (99000) | 6,200 | > 17,200 |
| KDM6B | 1 | 8-OH Quinoline (6400) | 6,200 | > 17,200 |
| KDM7B | 2.5 | 8-OH Quinoline (13000) | 400 | > 1,110 |

[a] The $IC_{50}$ values of positive control compounds are shown in parentheses.

[b] The $IC_{50}$ values for KDM enzymes represent at least n = 2 experiments. The $IC_{50}$ values of KDM5A and KDM5B were significantly lower than the enzyme concentration, indicating they likely underestimate compound potency since the assay may approach its lower limit.

[c] Fold selectivity is defined by $IC_{50}$ of KDM enzyme over KDM5A.

extracellular DNA, and secreted antigens by GS-5801 in PHH was due to a decrease in cccDNA levels. PHH were infected with HBV (genotype D; GTD) and cccDNA was established for three days prior to initiation of GS-5801 dosing with compound replenishment every three to four days for 14 days. Examination of cccDNA levels by Southern blot in PHH treated with GS-5801 indicated that GS-5801 did not alter levels of cccDNA out to 14 days of compound treatment (**S1 Fig in** S1 File). Thus GS-5801-mediated inhibition of intracellular HBV RNA and antigen secretion is not due to a reduction in levels of cccDNA but rather its impact on transcription of viral or host genes.

## GS-5801 exhibits antiviral activity across HBV genotypes

HBV has been classified phylogenetically into nine major genotypes, A–J, that exhibit between 4–8% nucleotide divergence as well as distinct geographical distributions [31]. To examine whether GS-5801 exhibited antiviral activity against HBV genotypes in addition to GTD, PHH were infected with patient sera from individuals infected with GTA, GTC, or GTE HBV genotypes that had previously been established as infectious in PHH. Following 14 days of dosing, GS-5801 reduced intracellular HBV RNA (median $EC_{50}$ = 1.1 μM, range across genotypes = 0.051–2.7 μM), extracellular HBV DNA (median $EC_{50}$ = 0.20 μM, range across genotypes = 0.079–0.25 μM), and secreted HBV antigens HBsAg ($EC_{50}$ = 0.17 μM, range across genotypes = 0.036–3.1 μM) and HBeAg (median $EC_{50}$ = 1.1 μM, range across genotypes = 0.30–1.4 μM; Table 2) in all genotypes examined.

## GS-080 is a potent and selective inhibitor of KDM5

The inhibitory activity of GS-080, the active parent of the prodrug GS-5801, was examined for all four members of the KDM5 family (KDM5A –D) by in vitro biochemical characterization. As summarized in Table 1, GS-080 had the highest inhibitory activity against KDM5A and KDM5B enzymes, with $IC_{50}$ values of 0.36 nM against KDM5A and 0.38 nM against KDM5B. To assess off-target effects of GS-080, the inhibitory activity of GS-080 was examined against other KDM enzymes including: KDM1, 2, 3, 4, 6, and 7 as well as a panel of HMT and HDAC enzymes (see **S3 Table in** S1 File). Measured GS-080 $IC_{50}$ values for all KDM enzymes were compared to the $IC_{50}$ value of GS-080 against KDM5A to calculate the fold selectivity for each KDM enzyme assayed (Table 1). GS-080 showed at least a 13-fold selectivity for KDM5A and KDM5B over members of the KDM4 family of enzymes, and a selectivity from > 1,100- to > 278,000-fold over members of the other KDM enzyme families tested. GS-080 showed no measurable inhibitory activity against any of the HMT or HDAC enzymes tested (**S3 Table in** S1 File; $IC_{50}$ values > 100 μM). Previously characterized KDM, HDAC, or HMT inhibitors were used as positive controls for the biochemical assay and included S2101,

**Table 2. Antiviral activity of GS-5801 in HBV-infected PHH.**

| PHH Donor[a] | Hu8181 | Hu8130 | Hu4167 | BCD | Hu7272 | Hu276 | Hu349 |
|---|---|---|---|---|---|---|---|
| vRNA $EC_{50}$ (μM) | 0.16 (1.9) | 1.1 (3.2) | 0.047 (1.1) | 0.034 (3.4) | 0.93 (3.7) | 0.042 (1.8) | 0.43 (1.2) |
| vDNA $EC_{50}$ (μM) | 0.14 (3.7) | 1.3[b] | 0.11 (3.5) | 0.0071 (4.9) | 1.1 (5.0) | 0.070 (1.2) | 0.23 (2.4) |
| HBsAg $EC_{50}$ (μM) | 0.061 (2.4) | 1.7 (3.7) | 0.24 (2.6) | 0.015 (3.3) | 5.1 (2.1) | 0.15 (1.8) | 0.97 (3.5) |
| HBeAg $EC_{50}$ (μM) | 0.11 (2.2) | 1.1 (2.3) | 0.15 (2.7) | 0.014 (19) | 0.61 (5.5) | 0.10 (2.0) | 0.38 (2.4) |
| $CC_{50}$ (μM) | > 10 | > 10 | > 10 | > 10 | > 10 | > 10 | > 10 |

[a] Data shown are the geometric mean $EC_{50}$ values and the geometric standard deviation factor from HBV-infected PHH treated every three to four days with GS-5801 for 14 days (n = 3 donor Hu8181, n = 3 donor Hu8130, n = 2 donor Hu4167, n = 2 donor BCD, n = 3 donor Hu7272, n = 2 Hu276, n = 2 Hu349 experiments).

[b] $EC_{50}$ value represents n = 1.

2,4-Pyridinedicarboxylic acid (PDCA), 8-hydroxy-5-quinolinecarboxylic acid (IOX1), and 8-hydroxyquinoline (8-OH Quinoline), which yielded $IC_{50}$ values consistent with literature [32, 33] (Table 1; **S3 Table in** S1 File).

## Depletion of *KDM5* by siRNA restricts HBV replication in PHH

To confirm that the antiviral activity we observed with the small molecule GS-5801 was due to inhibition of KDM5 in PHH, we examined the effect of depleting *KDM5* transcripts by RNA interference on HBV RNA and antigen production. Three days post HBV infection, *KDM5* transcripts were depleted with small interfering RNA (siRNA) two times during the infection time course to maintain transcript knockdown: once on Day 0 (three days post-infection) and again on Day 6 (nine days post-infection). *KDM5* transcripts were depleted with siRNA either individually to reduce levels of a single *KDM5* (*KDM5A*, *B*, *C*, or *D*) transcript or pooled to reduce levels of *KDM5A –D* transcripts in the cell. Thirteen days after initiation of siRNA treatment, amounts of *KDM5* transcripts as well as HBV RNA along with secreted antigens were measured by qRT-PCR or immunoassay, respectively. siRNA knockdown of *KDM5* transcripts individually (*KDM5A*, *B*, *C*, or *D*) or simultaneously (*KDM5A –D*) resulted in 55–78% inhibition of *KDM5* transcript levels in PHH (**S2A, S2B Fig in** S1 File) with no effect on cell viability as assessed by alamarBlue staining (**S2D Fig in** S1 File). Knockdown of *KDM5A*, *KDM5B*, *KDM5C*, or *KDM5D* transcripts individually in PHH resulted in mild repression of HBV RNA (26–43%), HBeAg (32–39%), and HBsAg (34–41%) by Day 13 after initiation of siRNA treatment (Fig 2). In contrast, simultaneous knockdown of *KDM5A*, *KDM5B*, *KDM5C*,

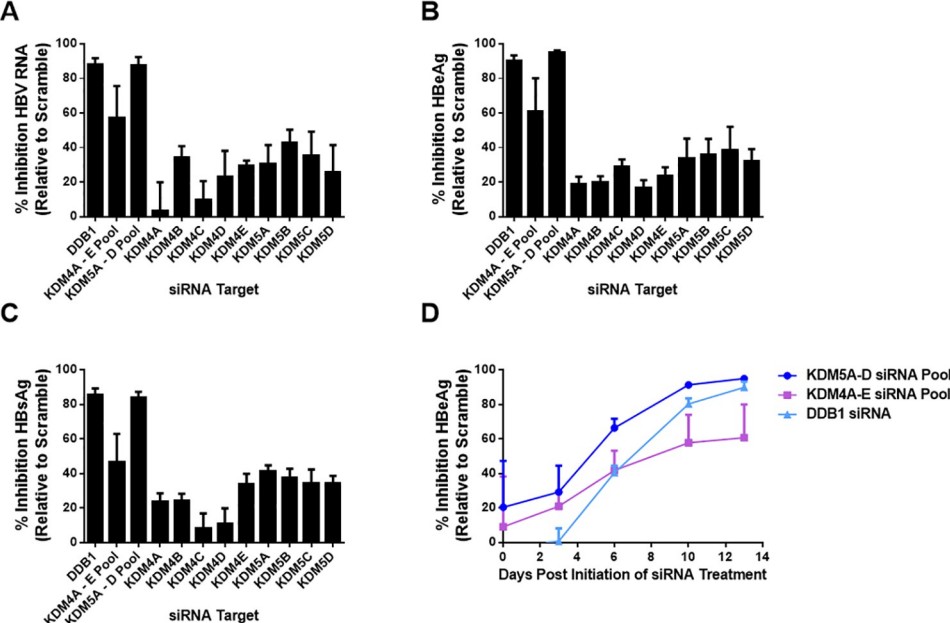

**Fig 2. Knockdown of *KDM5* confers antiviral activity in PHH.** *KDM4* and *KDM5* transcripts were depleted by siRNA either individually (*KDM4A*, *B*, *C*, *D*, or *E*; *KDM5A*, *B*, *C*, or *D*) or simultaneously (*KDM4A –E* pool; *KDM5A – D* pool) in HBV-infected PHH. *DDB1*, the well-characterized HBV host restriction factor, was also depleted by siRNA for comparison. Fourteen days after initiation of siRNA transfection, levels of (A) intracellular HBV RNA, (B) secreted HBeAg, and (C) secreted HBsAg were measured by qRT-PCR or immunoassay, respectively, for each siRNA condition. (D) The kinetics of HBeAg inhibition is shown for *KDM4A –E* pool, *KDM5A –D* pool, and *DDB1* siRNA conditions. Data shown are the average of two biological replicate experiments and error bars represent the standard deviation.

and *KDM5D* transcripts in PHH resulted in much greater repression of HBV RNA (88%), HBeAg (95%), and HBsAg (84%) by Day 13 (Fig 2). Knockdown of *KDM5A –D* transcripts in PHH resulted in similar inhibition of HBV replication as knockdown of the well-characterized host restriction factor DDB1; knockdown of *DDB1* in PHH yielded a reduction in HBV RNA (88%), HBeAg (91%), and HBsAg (86%) by Day 13 as expected [34–37] (Fig 2A–2C). The kinetics of antiviral activity in *KDM5*-depleted PHH, as measured by inhibition of HBeAg levels over time, suggested that antiviral activity was delayed after siRNA treatment with near maximal antiviral activity achieved by Day 10–13 post-initiation of siRNA treatment (Fig 2D). Together these data suggest that GS-5801 targets KDM5 to cause antiviral activity and that inhibition of all *KDM5* gene products (*KDM5A –D*) is likely required for antiviral activity.

Biochemical characterization of GS-080 indicates that it exhibits some inhibitory activity against the KDM4 family of histone lysine demethylases ($\geq$ 13-fold selectivity for KDM5A over KDM4A –C). To examine whether inhibition of KDM4 could also contribute to the antiviral effect observed with GS-5801 in PHH, levels of *KDM4* transcripts were knocked down individually (*KDM4A*, *B*, *C*, *D*, or *E*) or simultaneously (*KDM4A –E*) in PHH. siRNA knockdown of *KDM4* transcripts individually or simultaneously (*KDM4A –E*) resulted in 47–89% inhibition of *KDM4* transcript levels in PHH (**S2A, S2C in** S1 File) with no effect on cell viability as measured by alamarBlue staining (**S2D Fig in** S1 File). Knockdown of *KDM4A –E* transcripts simultaneously in HBV-infected PHH resulted in mild inhibition of HBV RNA (57%), HBeAg (61%), and HBsAg (47%) by Day 13 after initiation of siRNA treatment (Fig 2); however, *KDM4* depletion did not have as great of an effect on HBV replication compared to *DDB1* or *KDM5A –D* depletion.

## Antiviral activity of GS-5801 exhibits delayed kinetics

To characterize the kinetics as well as duration of GS-5801 antiviral activity in PHH, we examined the inhibition of secreted HBsAg and HBeAg by GS-5801 in HBV-infected PHH over time. PHH from three different donors were treated with GS-5801 by replacing drug-containing cell culture medium every three to four days for 30–32 days. Prior to re-treating cells with GS-5801, levels of HBsAg and HBeAg were measured by immunoassay from PHH supernatant. As shown in Fig 3, maximal HBsAg and HBeAg inhibition by GS-5801 was achieved by 12–17 days after initiation of treatment; similar to the antiviral kinetics observed with *KDM5A –D* siRNA treatment (Fig 2D). GS-5801 antiviral activity was maintained throughout the experimental time course of 30–32 days (Fig 3), indicating that treatment of PHH with GS-5801 maintains HBV antigen reduction in vitro.

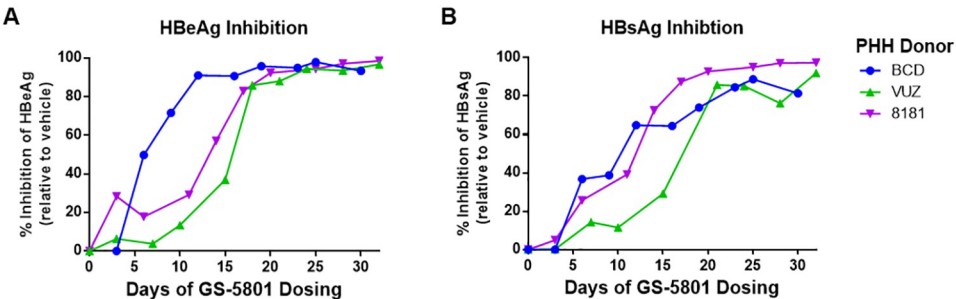

**Fig 3. Antiviral activity of GS-5801 exhibits delayed kinetics.** PHH from three donors (BCD, VUZ, and 8181) were infected with HBV for three days prior to initiation of GS-5801 treatment. PHH were dosed with vehicle or 10 μM GS-5801 every three to four days for a total of 30–32 days. At the timepoints shown, levels of secreted (A) HBeAg and (B) HBsAg were measured by immunoassay. Data are plotted as the percentage inhibition of HBeAg or HBsAg in GS-5801 treated PHH relative to vehicle treated PHH.

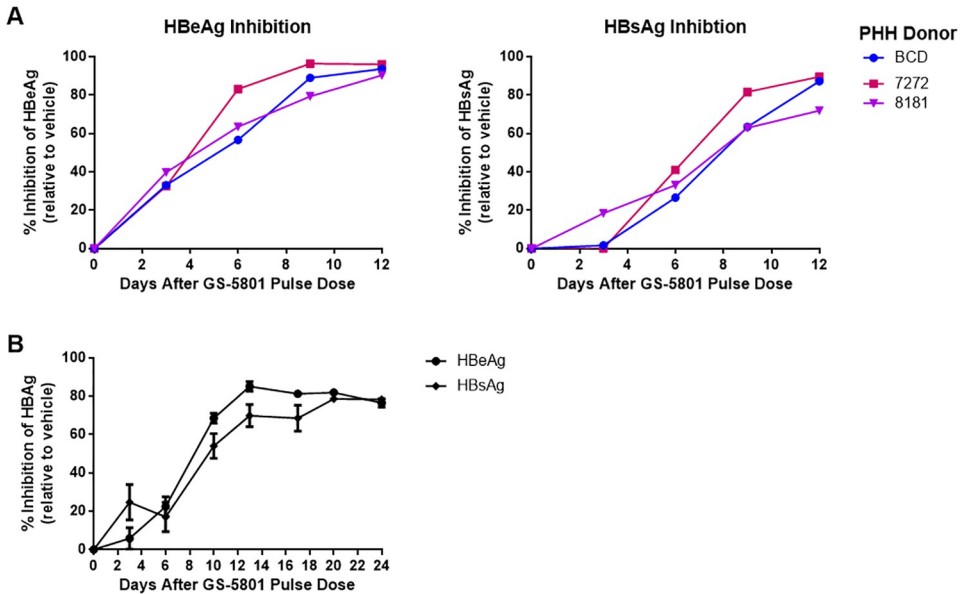

**Fig 4. A single dose of GS-5801 confers sustained HBV antiviral activity in PHH.** (A) PHH from three donors (BCD, 7272, and 8181) were infected with HBV for three days prior to initiation of GS-5801 treatment. PHH were treated once for two hours with vehicle or 10 μM GS-5801. Compound was removed by replacing the medium with fresh medium without drug or vehicle. At the timepoints shown, levels of secreted HBeAg and HBsAg were measured by immunoassay for 12 days. Data are shown as the percentage inhibition of HBeAg or HBsAg in GS-5801 treated PHH relative to vehicle treated PHH. (B) PHH from donor BCD were infected with HBV for three days prior to initiation of GS-5801 treatment. PHH were treated once for two hours with vehicle or 10 μM GS-5801. Compound was removed by replacing the medium with fresh medium without drug or vehicle. At the timepoints shown, levels of secreted HBeAg and HBsAg were measured by immunoassay for 24 days. Data are shown as the average percentage.

## GS-5801 causes sustained HBV antigen suppression in PHH

To investigate whether continuous treatment with GS-5801 is necessary to achieve a sustained antiviral response in HBV-infected PHH, we treated PHH with a single, two-hour pulse dose of GS-5801 before removing compound. Antiviral activity was monitored over time by measuring the levels of secreted HBeAg and HBsAg. A pulse dose of GS-5801 reduced cell-associated active parent GS-080 exposure in PHH compared to continuous dosing (**S3 Fig in** S1 File; a single two-hour pulse dose of GS-5801 resulted in two-fold lower $C_{max}$ and $AUC_{last}$ values compared to continuous dosing). Furthermore, a pulse dose of GS-5801 in HBV-infected PHH from three independent donors was sufficient to confer antiviral activity with similar kinetics to continuous dosing of GS-5801 (Figs 3 and 4A). Longer time course experiments probing the antiviral activity of GS-5801 demonstrated that inhibition of HBeAg and HBsAg was sustained up to 20 days following a pulse dose (Fig 4B).

## Global H3K4me3:H3 increases correlate with GS-5801 antiviral activity in PHH

It has been demonstrated that inhibition of KDM5 either genetically or pharmacologically increases levels of H3K4me3 on the mammalian genome [24, 27]. Therefore, we investigated the relationship between global H3K4me3 changes and antiviral activity in HBV-infected PHH treated with GS-5801. HBV-infected PHH were treated either continuously with GS-5801 every three to four days or once with a two-hour pulse dose of GS-5801and the ratio of the levels of H3K4me3 relative to H3 (H3K4me3:H3) were measured by ELISA along with

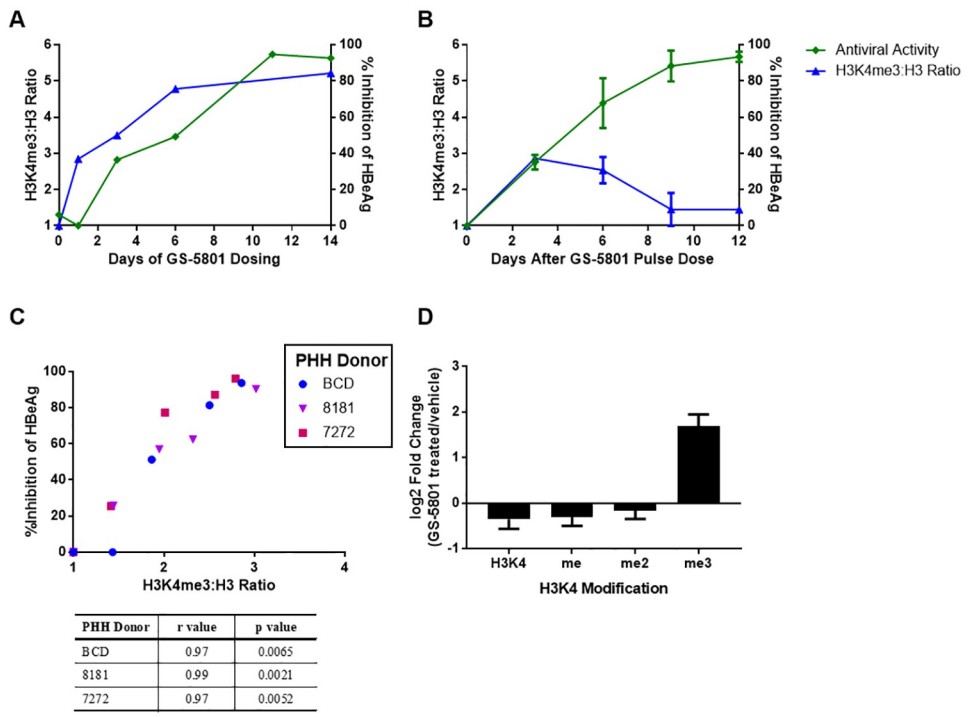

**Fig 5. GS-5801 causes global increases in H3K4me3:H3 that precede antiviral activity.** PHH were infected with HBV for three days prior to initiation of GS-5801 treatment. (A) PHH were treated with vehicle or 10 μM GS-5801 every three to four days for a total of 14 days (continuous dosing). Data shown are from donor 8130 and are representative of data from continuous dosing experiments. (B) PHH were treated with vehicle or 10 μM GS-5801 once for two hours with vehicle or 10 μM GS-5801 prior to replacing medium with fresh medium without drug or vehicle (pulse dosing). Note that the half-life of active parent of GS-5801 in hepatocytes is 42–66 hrs leading to sustained exposure during the 3 day periods. Data shown are the average values from three PHH donors (BCD, 8181, 7272) and error bars represent the standard deviation. (A, B) At the timepoints shown, levels of secreted HBeAg were measured by immunoassay and are plotted as the percentage inhibition of HBeAg in GS-5801 treated PHH relative to vehicle treated PHH. In addition, levels of H3K4me3 relative to H3 (H3K4me3:H3) were measured by ELISA in vehicle and GS-5801 treated PHH and are plotted as the ratio of H3K4me3:H3 measured in GS-5801 treated PHH relative to vehicle treated PHH. (C) The relationship of H3K4me3:H3 ratio increases to HBeAg inhibition in PHH (donors BCD, 8181, 7272) treated with a single two hour pulse dose of 0.1, 1, 3, or 10 μM GS-5801 relative to vehicle-treated PHH are shown. The H3K4me3:H3 ratio of GS-5801 treated cells was normalized to vehicle treated cells at day three after a two hour pulse dose. The percentage inhibition of HBeAg was measured at the end of the experiment (day 12 or 14 post-initiation of dosing) and plotted as the inhibition of HBeAg in GS-5801 treated PHH relative to vehicle treated PHH. (D) Uninfected PHH cells treated with vehicle or 10 μM GS-5801 every 3 days. On Day 7 (24 hours after last GS-5801 dose) histones were purified and levels of H3K4 (unmodified), H3K4me1 (methylated), H3K4me2 (di-methylated), and H3K4me3 (trimethylated) TKQTAK peptides were measured by mass spectrometry. Data shown are the average ratios of unmodified, methyl, dimethyl, and trimethyl H3K4 peptide TKQTAR (log2 (GS-5801 treated/vehicle)) measured from three biological replicates. Error bars represent standard deviation.

levels of secreted HBeAg to monitor antiviral activity. Continuous dosing of HBV-infected PHH with GS-5801 resulted in increased levels of the global H3K4me3:H3 ratio throughout the experimental time course with an apparent saturation of H3K4me3:H3 (4.8-fold increase compared to vehicle treated PHH) occurring six days after initiation of dosing (PHH received two doses of GS-5801; Fig 5A). In contrast, a single pulse dose of GS-5801 resulted in a 2.8-fold increase in H3K4me3:H3 by day three that decreased to near baseline levels by day nine (Fig 5B). This temporary increase in the H3K4me3:H3 ratio peaking at day three seems to be sufficient for an gradual increase in inhibition of HBeAg as indicated by the antiviral effect on day 12. Thus, increases in H3K4me3:H3 after a pulse dose of GS-5801 precedes the maximal antiviral response. Having established the delayed effect of s pulse dose, we tried to

determine the relationship between the H3K4me3:H3 ratio and the maximal HBeAg inhibtion by using a short pulse treatment with increasing concentration of GS-5801 (two hours on day 1 with a concentration range of 0.016–10 μM; Fig 5C). A near 100% inhibtion level on day 12 is associated with a H3K4me3:H3 ratio of ~3, and that can be reached with a short pulse. . . Quantitative mass spectrometry analysis of histones purified from GS-5801 treated PHH corroborated that GS-5801 predominantly caused an increase in trimethylated H3K4 (H3K4me3) as opposed to dimethyl (H3K4me2) or monomethyl (H3K4me) species in PHH (Fig 5D). Together these data suggest that levels of the H3K4me3:H3 ratio positively correlate with HBV antiviral activity after GS-5801 treatment and that sustained increases in H3K4me3:H3 are not required for GS-5801 antiviral activity. Note that this observation was very intriguing as it suggests that GS-5801 could be developed clinically with infrequent dosing, potentially just one weekly dose.

## GS-5801 alters the expression of viral as well as host transcripts in PHH

Given that GS-5801 increases H3K4me3:H3, which is an epigenetic modification associated with transcriptionally active promoters [18–22], we examined the effect of GS-5801 on host and viral transcriptomes using RNA sequencing (RNA-seq). HBV-infected PHH from three donors were treated with GS-5801 every three to four days for 13 days and total cellular mRNA was isolated and sequenced on days 1, 3, 10, and 13 after initiation of dosing. Corroborating our qRT-PCR data (see Table 2), levels of HBV mRNA as measured by RNA-seq were decreased after GS-5801 treatment in a dose and time dependent manner (Fig 6A). Global analyses of host gene expression changes upon GS-5801 treatment (defined as transcripts up or down regulated by $\geq$ 4-fold compared to vehicle treated PHH, FDR < 0.05) revealed numerous dose and time dependent effects on host transcripts (Fig 6B; **S6 Table in** S1 File), with most GS-5801-regulated genes sustained in their expression pattern over the time course (Fig 6C). The majority of genes that changed in response to GS-5801 treatment were upregulated and included genes that spanned a large number of pathways including those related to cytoskeleton remodeling, cell-cell junction organization, cytokine signaling and DNA repair (Fig 6D) However, by 10 days after treatment genes were also observed to be downregulated by GS-5801 treatment and included genes involved in pathways such as metabolism and biosynthesis (**S6 Table in** S1 File). Notably, interferon-α-stimulated genes (ISG) [38] showed mild transcriptional regulation by GS-5801, but as a class did not exhibit a strong pattern of differential regulation in response to GS-5801 treatment of PHH (**S4 Fig in** S1 File). We concluded that the vastness of transcriptome changes and the heterogeneity seen in response to GS-5801 would make it extremely difficult to find a causative link to the antiviral effect.

## GS-5801 is liver-targeted in nonclinical species and preferentially increases H3K4me3:H3 levels in the liver

As HBV is a liver-tropic virus, our pro-drug strategy sought to not only improve the cell-permeability of GS-080 but also its enrichment in the liver. Liver enrichment of GS-080 could reduce systemic exposure to limit epigenetic alteration of host chromatin in other tissues. GS-5801 was identified as an ethyl ester pro-drug of GS-080 with high liver to plasma area under the curve (AUC) ratios for the active parent GS-080 in cynomolgus monkey (Fig 7A; 176-fold GS-080 liver to plasma AUC ratio) and rat (Fig 7B; 44-fold GS-080 liver to plasma AUC ratio). To investigate the effect of preferential liver exposure of GS-080 on the pharmacodynamic response in vivo (H3K4me3:H3), we dosed cynomolgus monkeys and rats orally once daily with GS-5801 for seven or five days, respectively, and examined global levels of the H3K4me3: H3 ratio by ELISA in liver, lung, kidney, and PBMCs 24 hours after the last dose of GS-5801.

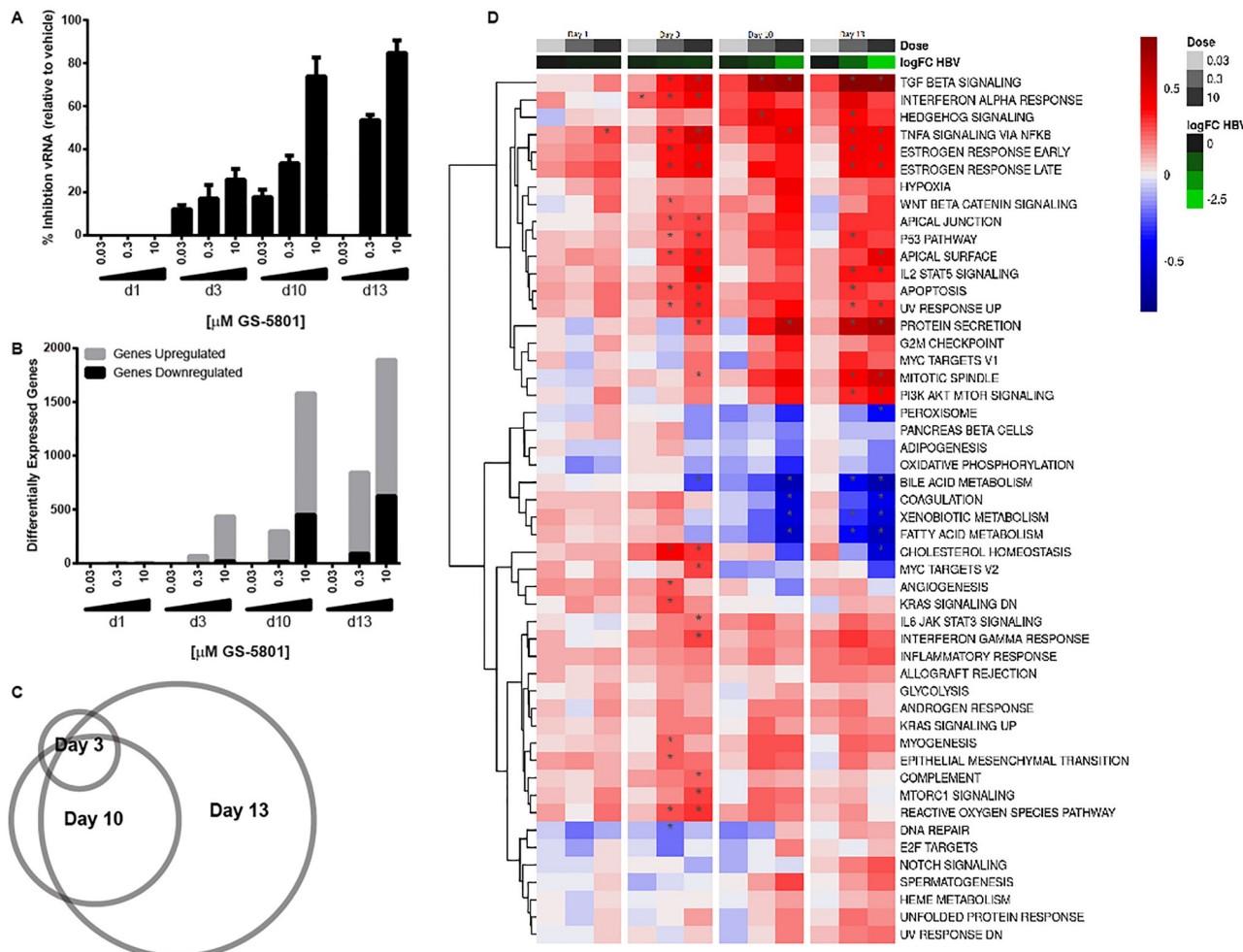

**Fig 6. GS-5801 influences host and viral gene transcription.** PHH from three donors (8130, 8181, and 4239) were infected with HBV for three days prior to initiation of GS-5801 treatment. PHH were treated continuously with vehicle, 0.03, 0.3, or 10 μM GS-5801 every three to four days for a total of 13 days. PHH were harvested at 1, 3, 10, and 13 days after initiation of GS-5801 dosing and mRNA was quantified by RNA-seq. (A) The number of sequencing reads mapping to the HBV genome were quantified in all samples (counts per million; cpm) and data are plotted as the percentage inhibition of HBV RNA (vRNA) in GS-5801 treated samples relative to vehicle treated samples at each timepoint. (B) The number of genes differentially up or downregulated (differential by 4-fold $\log_2$; FDR < 0.05) in GS-5801 treated PHH compared to vehicle treated PHH are shown for each timepoint. (C) Shown is a Venn diagram demonstrating the overlap of the identity of genes differentially upregulated (fold change ≥ 4-fold $\log_2$; FDR < 0.05) in PHH treated with 10 μM GS-5801 compared to vehicle treated PHH for Day 3 (n = 414 genes), Day 10 (n = 1131 genes), and Day13 (n = 1268 genes) after initiation of GS-5801 dosing. (D) Hallmark Gene Set. Data are displayed as the difference in average gene set score based on GSVA between each dose respective vehicle control, with upregulated gene sets in red and downregulated gene sets in blue. For comparison, downregulation of HBV mRNA for same dose and time regiment is shown at the top.

Treatment of monkeys (Fig 7C) or rats (Fig 7D) with GS-5801 resulted in increased levels of the H3K4me3:H3 ratio in the liver compared to lung, kidney, or PBMCs at most doses tested. Furthermore, no adverse effects were observed in rats or monkeys at the GS-5801 doses tested in these studies. Together these data indicate that GS-5801 is a liver-targeted pro-drug that preferentially causes accumulation of H3K4me3:H3 on total cellular DNA at doses that are well tolerated in vivo.

To examine whether increases in H3K4me3:H3 ratios are reversible, similar to the recovery of H3K4me3:H3 ratios to near baseline levels in PHH after a pulse dose of GS-5801 (see Fig 5B), rats were dosed once daily for seven days with 0, 10, 30, or 100 mg/kg GS-5801 and a

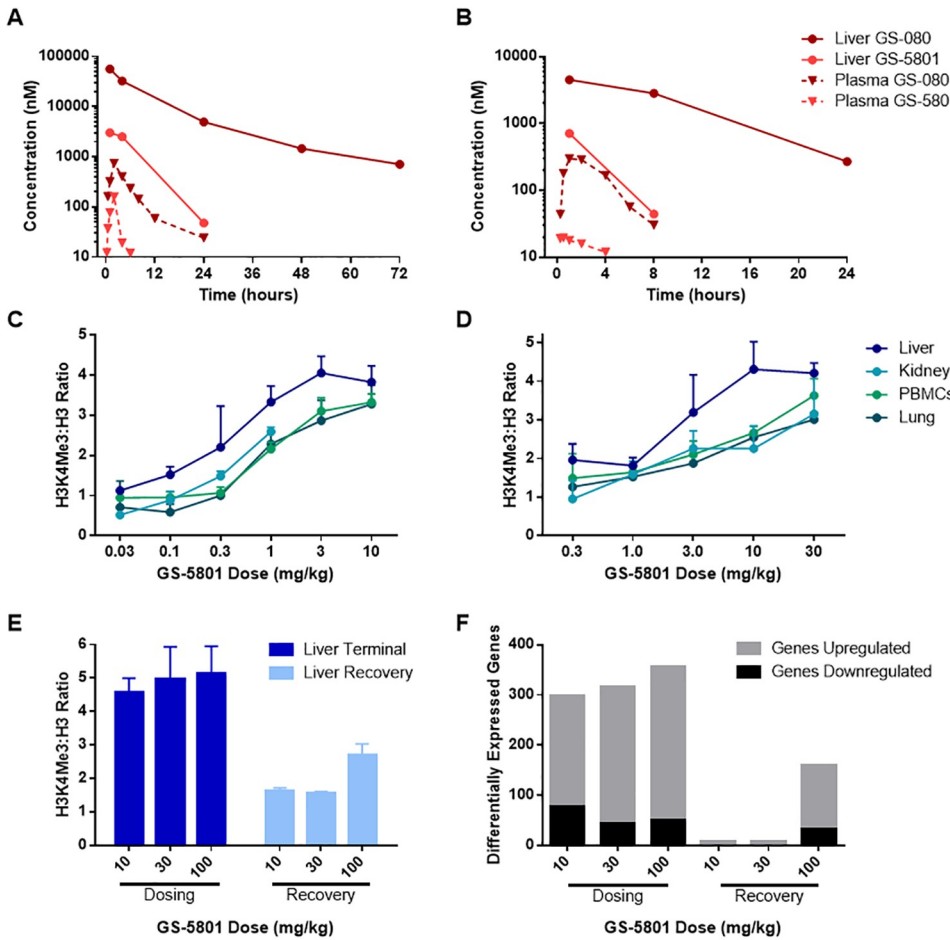

**Fig 7. GS-5801 increases H3K4me3:H3 levels preferentially in liver tissue.** Sprague Dawley rats (A) and cynomolgus monkeys (B) were dosed once with 1 mg/kg or 2.5 mg/kg GS-5801 p.o., respectively. Amounts of pro-drug (GS-5801) and active parent (GS-080) were measured by LC-MS in liver tissue and plasma at the timepoints shown. (C) Cynomolgus monkeys were dosed p.o. once daily for seven days with 0.03, 0.1, 0.3, 1, 3, or 10 mg/kg of GS-5801. Twenty four hours after the last dose, amounts of H3K4me3 and H3 were measured by ELISA in liver, kidney, PBMCs, and lung. Data are displayed as the average H3K4me3:H3 ratio of GS-5801 treated monkeys relative to vehicle treated monkeys from n = 2–3 animals; error bars represent the standard deviation. (D) Sprague Dawley rats were dosed p.o. once daily for five days with 0.3, 1, 3, 10, or 30 mg/kg of GS-5801. Twentyfour hours after the last dose, amounts of H3K4me3 and H3 were measured by ELISA in liver, kidney, PBMCs, and lung. Data are displayed as the average H3K4me3:H3 ratio of GS-5801 treated rats relative to vehicle treated rats from n = 3 animals; error bars represent the standard deviation. (E–F) Wistar Han rats were dosed p.o. once daily for seven days with 10, 30, or 100 mg/kg of GS-5801 and a subset of rats from each dose group continued into a seven day off-treatment phase. (E) Twentyfour hours after the last dose or recovery day, amounts of H3K4me3 and H3 were measured by ELISA in the liver. Data are displayed as the average H3K4me3:H3 ratio of GS-5801-treated rats relative to vehicle treated rats from n = 3 animals during the dosing or recovery phases; error bars represent the standard deviation. (F) Transcript levels in rat liver tissue were quantified by RNA-seq (cpm). The number of genes differentially up or downregulated (differential by 2-fold $\log_2$; FDR < 0.05) in GS-5801-dosed rats compared to vehicle-treated rats are shown during the dosing phase (24 hours after once daily dosing for seven days) and recovery phase (24 hours after seven days off drug; F).

subset of animals continued for seven days off-treatment (recovery phase). Levels of H3K4me3:H3 in rat liver were evaluated by ELISA after seven days of once daily dosing (dosing phase) as well as after seven days off treatment (recovery phase). H3K4me3 and H3 ELISA measurements demonstrated increases in H3K4me3:H3 after seven days of GS-5801 once daily dosing compared to vehicle-treated animals (4.6- to 5.2-fold increase; Fig 7E) that agreed with H3K4me3:H3 increases previously observed when rats were dosed once daily for five

days with 10 or 30 mg/kg GS-5801 (4.2- to 4.3-fold; see Fig 7D). When rats were taken off treatment for seven days, H3K4me3:H3 ratio levels returned to near vehicle levels in the 10 and 30 mg/kg dose groups (≤ 1.6-fold of vehicle animals; Fig 7E); however, H3K4me3:H3 ratio levels in the 100 mg/kg dose group were still elevated above vehicle (2.7-fold increase compared to vehicle treated animals; Fig 7E). Next, we examined whether there were any corresponding changes in transcript expression levels that paralleled H3K4me3:H3 increases in GS-5801 treated rats after seven days of once daily treatment as well as after seven days off treatment. Total mRNA from liver tissue of rats dosed for seven days with GS-5801 (0, 10, 30, or 100 mg/kg GS-5801) as well as rats taken off drug for seven days was sequenced with RNA-seq. The number of differentially expressed transcripts were similar after seven days of once daily GS-5801 dosing in rat liver among the 10, 30, and 100 mg/kg dose groups with approximately 300 transcripts differentially regulated between GS-5801 treated groups and vehicle treated (≥ 2-fold differentially regulated, FDR < 0.05; Fig 7F). The recovery of H3K4me3:H3 ratios in GS-5801 treated animals to H3K4me3:H3 ratios in vehicle treated animals after seven days off treatment (see Fig 7E) corresponded to a return of transcript levels to near vehicle treated levels (Fig 7F).

## GS-5801 is not efficacious in a humanized mouse model of HBV infection

To determine whether the antiviral activity of GS-5801 we observed in PHH translated to antiviral efficacy in a nonclinical model, we infected urokinase-type plasminogen activator (uPA) severe combined immunodeficiency (SCID) mice with humanized livers [39] with HBV (genotype C, GTC). HBV infected mice were dosed with 30 or 100 mg/kg of GS-5801. These doses were selected to cause increases in liver H3K4me3:H3 ratios that were comparable to those seen in the PHH infection model that were associated with HBV antiviral activity. Since GS-5801 demonstrated increased levels of H3K4me3:H3 ratios in rat and monkey liver tissue that was well-tolerated and reversible after a seven-day recovery period (see Fig 7), we selected a dosing schedule of once daily oral dosing for one week on treatment (qd x 7d) followed by one week off treatment for a total of 56 days (4 treatment cycles). Every seven days, levels of serum HBV DNA and HBsAg were measured to assess the antiviral activity of GS-5801 in this HBV infection model. Treatment of HBV-infected humanized mice with GS-5801 did not result in changes in the amounts of HBsAg or HBV DNA during any of the dosing cycles or recovery periods (Fig 8A and 8B). At the end of the study (Day 56), we measured H3K4me3 and H3 levels in mouse liver tissue by ELISA and observed 2.9- and 3.3-fold increases in H3K4me3:H3 ratios in mice dosed with 30 or 100 mg/kg of GS-5801, respectively, compared to vehicle-treated animals. The increase in H3K4me3:H3 ratios in mouse livers corresponded to an increase in H3K4me3:H3 that was expected to result in antiviral activity for GS-5801 as was observed in the PHH infection model (see Fig 5). Furthermore, the antiviral activity of GS-5801 was assessed in vitro using the primary human hepatocytes that reconstituted the uPA-SCID mouse livers and the HBV GTC virus used in the murine efficacy model. Treatment of these GTC HBV-infected hepatocytes in vitro with GS-5801 every 3–4 days resulted in a reduction of HBsAg and HBeAg levels (Fig 8D) similar to what was observed previously with multiple PHH donors and HBV viruses (see Fig 5, Tables 2 and 3).

## Discussion

GS-5801 is an ethyl ester prodrug that metabolizes to GS-080, a potent and selective inhibitor of KDM5a-d. KDM5 enzymes are epigenetic modifiers that demethylate lysine 4 of histone 3 in nucleosomes to regulate gene transcription [14, 15]. In a PHH model of HBV infection, GS-5801 demonstrates antiviral activity across multiple HBV genotypes by reducing HBV RNA,

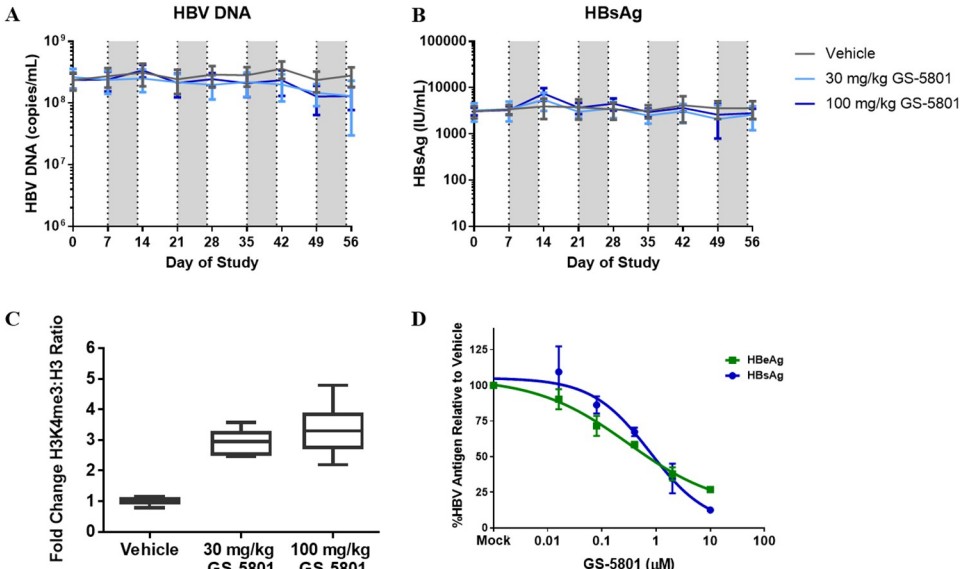

**Fig 8. GS-5801 does not reduce HBV DNA or HBsAg levels in a humanized mouse model of HBV infection.**
uPA-SCID mice with humanized livers were infected with HBV GTC virus and dosed p.o. with 30 or 100 mg/kg of GS-5801once daily for seven days in a one week on one week off dosing regimen. Every seven days, levels of HBV DNA (A) and HBsAg (B) were measured. Data is plotted as the average of n = 6–8 mice/timepoint/dose group, error bars represent the standard deviation. Shaded grey areas indicate the dosing periods. (C) At study endpoint, Day 56, levels of H3K4me3 and H3 were measured in liver tissue via ELISA to assess GS-5801 pharmacodynamics. Data is shown as the average fold change in H3K4me3:H3 ratio in mice (n = 6–8 per dose group) treated with 30 mg/kg or 100 mg/kg of GS-5801 and normalized to vehicle-treated animals; error bars represent the standard deviation. (D) The antiviral activity of GS-5801 was assessed in vitro with the same hepatocytes and GTC virus used in the in vivo HBV efficacy model. Hepatocytes were plated and infected with HBV prior to GS-5801 compound treatment. At Day 21, antiviral activity was assessed by measuring extracellular levels of HBsAg and HBeAg. Data is plotted as the average of biological duplicate samples; error bars represent the standard deviation.

DNA, and antigen levels without altering amounts of cccDNA, the transcriptional template of HBV. Knockdown of *KDM5* mRNA levels with siRNA cause a decrease in HBV RNA, DNA, and antigen levels, which further corroborated the role of KDM5 in HBV replication. GS-5801 causes a sustained reduction in HBV replication in PHH that correlates with increases in global cellular H3K4me3:H3 levels. Transient increases in H3K4me3:H3 are sufficient for GS-5801 antiviral activity in HBV-infected PHH, suggesting that an in vivo dosing strategy of finite treatment periods could be efficacious.

Detailed in vivo characterization of GS-5801 pharmacokinetic and pharmacodynamic properties (H3K4me3:H3 ratio) reveal that GS-5801 loads the liver with active drug (GS-080)

**Table 3. Antiviral activity of GS-5801 across HBV genotypes.**

| EC$_{50}$ (μM)[a] | GTA | GTC | GTD | GTE |
|---|---|---|---|---|
| vRNA | 1.8 (1.7) | 0.051 (1.7) | 0.43 (1.5) | 2.7 (3.2) |
| vDNA | 0.17 (2.2) | 0.079 (2.0) | 0.22 (1.6) | 0.25 (7.1) |
| HBsAg | 0.036 (1.0) | 0.12 (10) | 0.22 (1.0) | 3.1 (17) |
| HBeAg | 1.1 (1.5) | N/A[b] | 0.30 (1.1) | 1.4 (2.4) |

[a] Data shown are the geometric mean EC$_{50}$ values and the geometric standard deviation factor from PHH infected with the indicated HBV genotype and treated every three to four days with GS-5801 for 13 days (n = 2 experiments).
[b] Data not available; HBeAg levels were below limit of quantitation by immunoassay.

in both nonclinical species examined: rat and cynomolgus monkey (44- and 176-fold liver to plasma AUC ratio, respectively). The enhanced liver levels of GS-080 in rats and cynomolgus monkeys correlates with higher H3K4me3:H3 ratios in the liver compared to other cells and tissues (PBMC, lung, or kidney). Since HBV is a liver-tropic virus that infects hepatocytes, enhancing the distribution of GS-080 to the liver could provide a wider therapeutic index.

Utilizing knowledge gained from our rat and monkey pharmacokinetic and pharmacodynamic in vivo models as well as our in vitro PHH infection model, we evaluated the antiviral activity of GS-5801 in a humanized mouse model of HBV infection. To our surprise, GS-5801 did not have antiviral activity at either dose tested (30 mg/kg and 100 mg/kg) in the humanized model of HBV infection despite causing H3K4me3:H3 ratios similar to those seen in PHH (2.9- and 3.3-fold increases) based on our in vitro PHH HBV infection data. The reason for the disconnect in GS-5801 antiviral activity in vivo and in vitro is unclear but indicates that increases in H3K4me3:H3 ratio in the liver of HBV-infected humanized mice do not predict antiviral activity as was seen for GS-5801 in the PHH model. Thus, the simple PD marker for KDM5 engagement, the H3K4me3:H3 ratio, cannot be considered a straightforward pharmacodynamic marker for GS-5801 antiviral efficacy.

Considering how important a new host target for HBV therapy would be, and that the HBV-infected humanized mouse model might not completely predict GS-5801 performance in humans, a small Phase1a/b study was carried out with healthy volunteers and HBV patients (details to be published elsewhere). To our disappointment, GS-5801 was associated with dose limiting, reversible liver toxicity at exposures below reaching significant H3K4me3:H3 ratio increases in patient PBMCs, consistent with the absence of any HBV antigen level decreases (to be published elsewhere), rendering our clinical study inconclusive. Nevertheless, any further clinical evaluation of KMD5 inhibitors in other therapeutic areas should pay close attention to hepatotoxic effects, which might me associated with inhibition of the biologic target even at low levels of engagement.

Nevertheless, the mechanism by which KDM5 inhibition leads to HBV antiviral activity in PHH remains to be elucidated but should serve to decipher the disconnect between GS-5801 antiviral activity in PHH and the humanized mouse model of HBV infection. Two non-mutually exclusive mechanisms of action for GS-5801 antiviral activity in PHH are hypothesized: (1) modulation of host transcripts by GS-5801 restricts HBV replication indirectly and (2) alteration to the epigenetic landscape of HBV cccDNA reduces HBV transcription directly. As demonstrated by transcriptional profiling of PHH, GS-5801 causes changes in expression levels of many host transcripts and modulation of one or more host transcripts could mediate GS-5801 antiviral activity. Indeed, KDM5 was demonstrated to play a role in respiratory syncytial virus (RSV) pathogenesis as chemical or genetic inhibition of KDM5 in human dendritic cells infected with RSV led to increased transcription of pro-inflammatory cytokines that inhibited RSV replication [40]. In the PHH HBV infection system, the many host transcriptional changes induced by GS-5801 in PHH could make these hepatocytes generally less permissive for HBV replication in vitro. This phenomenon has been proposed for the anti-HBV activity of the retinoid class of compounds which displayed potent antiviral activity in PHH that, however, did not translate in the same humanized mouse model of HBV infection we have employed for GS-5801 [13]. For the retinoid Accutane, its lack of in vivo antiviral activity was hypothesized to be related to differences in transcriptional regulation observed in PHH in vitro versus liver tissue in vivo. Overall, far fewer transcripts were regulated by Accutane in vivo (humanized mouse liver) compared to in vitro (PHH). It was further shown that the culturing process of PHH required to establish permissibility for HBV infection leads to transcriptome changes, many of which were reversed by retinoids [13]. Thus, in vitro transcriptional changes do not always predict in vivo transcriptional regulation. For GS-5801

we observed a similar phenomenon in its in vitro versus in vivo transcriptional regulation, with many fewer host transcripts modulated in liver tissue in vivo (Fig 7F) compared to PHH in vitro (Fig 6B), despite achieving similar increases of H3K4me3:H3. It is clear that similar H3K4me3:H3 changes cannot be tied to similar transcriptional responses, thus, rendering our pharmacodynamic marker a poor predictor of GS-5801 in vivo activity.

Our second hypothesis for GS-5801 antiviral activity in vitro is that inhibition of KDM5 by GS-5801 directly alters the epigenetic landscape of HBV cccDNA to reduce HBV transcription. Although as the consequences of H3K4 trimethylation on cccDNA are not well characterized, activating PTMs (e.g. H3K4me3, H3K27ac) have been associated with HBV transcriptional start sites suggesting that, similar to the host, chromatin modifications contribute to the transcriptional regulation of cccDNA [8–11]. Thus GS-5801 could inhibit HBV transcription by decreasing H3K4me3 levels or changing the spatial organization of H3K4me3 on HBV cccDNA chromatin. This a phenomenon that has been described for some bivalent promoters in eukaryotic cells as the reduction of gene transcription has been correlated with H3K4me3 spread into gene bodies [24]. Alternatively, GS-5801 could also serve to modulate the epigenetic landscape of cccDNA by altering epigenetic modifications that repress transcription of cccDNA (e.g. through H3K27me3 or H3K9me3 that are associated with transcriptionally repressed genes on eukaryotic chromatin) [12, 41].

In conclusion, this study details nonclinical work characterizing the antiviral activity and pharmacodynamic effects of a small molecule inhibitor of KDM5. GS-5801 demonstrates antiviral activity in a PHH model of HBV infection that correlates with increases in global cellular H3K4me3:H3 ratio, but no antiviral activity is seen in the humanized mouse model of HBV infection despite reaching the desired pharmacodynamic effects expected to be efficacious against HBV. This work highlights the difficulty of epigenetic approaches for therapeutic intervention, especially when the pharmacodynamic effects do not indicate the required engagement of the actual downstream pathogen target(s).

## Materials and methods

### Ethics statement

Primary human hepatocytes (PHH) isolated from deceased donor livers were purchased from Thermo Fisher Scientific (Waltham, MA), Lonza (Basel, Switzerland), BioreclamationIVT (Westbury, NY), and Corning, Inc. (Corning, NY). Consent was obtained from the donor or the donor's legal next of kin for use of the tissue and its derivatives for research purposes using IRB-approved authorizations. Plasma from CHB patients was purchased from Proteogenex (Culver City, CA) or BioCollections Worldwide, Inc (Miami, FL). Consent was obtained from the donor for use of the sample for research purposes using IRB-approved authorizations. All animal work was performed by Covance, Inc. (Princeton, NJ), Crown BioScience, Inc. (Santa Clara, CA) or PhoenixBio Inc. (Higashi-Hiroshima, Japan). Studies in nonclinical species were conducted at test sites fully accredited by the Association for Assessment and Accreditation of Laboratory Animal Care (AAALAC). All procedures in the protocol were in compliance with applicable animal welfare acts and were approved by the local Institutional Animal Care and Use Committee (IACUC) or Animal Ethics Committee. An attending laboratory veterinarian was responsible for providing the medical treatment necessary to prevent unacceptable pain and suffering for the animals on study. All surgery was performed under isoflurane anesthesia, and all efforts were made to minimize suffering.

For the clinical studies mentioned above, all patients signed an informed consent form before screening and in accordance with local regulatory and ethics committee requirements. The experimental protocol in these trials was approved by Gilead Sciences and all local

regulatory agencies (ANZCTR: **A Phase 1b Study Evaluating the Safety and Tolerability of GS-5801 in Patients with Chronic Hepatitis B (ACTRN12616001375448)**).

## Compounds

GS-5801, GS-080, GS-420 and GS-444 were synthesized by Gilead Sciences, Inc., CanAm Bioresearch, Inc. (Winnipeg, Canada) or by Shanghai Medicilon, Inc. (Shanghai, China). For cell-based assays, compounds were formulated in 100% DMSO at a concentration of 10 mM. For in vivo studies, GS-5801 was formulated in deionized water.

**GS-5801 (isolated as the *bis*-tosylate salt).** $^1$H NMR (400 MHz, Methanol-$d_4$) δ 8.79 (dd, $J$ = 5.1, 0.8 Hz, 1H), 7.95 (s, 1H), 7.90 (dd, $J$ = 5.1, 1.5 Hz, 1H), 7.68 (d, $J$ = 8.2 Hz, 4H), 7.23 (d, $J$ = 7.9 Hz, 4H), 4.54 (s, 2H), 4.43 (q, $J$ = 7.1 Hz, 2H), 4.32 (s, 2H), 3.82 (t, $J$ = 5.8 Hz, 2H), 3.46–3.34 (m, 4H), 2.98 (s, 6H), 2.37 (s, 6H), 1.41 (t, $J$ = 7.1 Hz, 3H), 1.24 (t, $J$ = 7.1 Hz, 3H). LCMS-ESI$^+$ (*m/z*): [M+H]$^+$ calcd 337.2; found 337.2.

**GS-080 (isolated as the *bis*-HCl salt).** $^1$H NMR (400 MHz, Methanol-$d_4$) δ 8.78 (dd, $J$ = 5.0, 0.7 Hz, 1H), 7.95 (s, 1H), 7.90 (dd, $J$ = 5.0, 1.4 Hz, 1H), 4.52 (s, 2H), 4.25 (s, 2H), 3.81 (t, $J$ = 6.2 Hz, 2H), 3.45–3.34 (m, 4H), 2.98 (s, 6H), 1.25 (t, $J$ = 7.1 Hz, 3H). LCMS-ESI$^+$ (*m/z*): [M+H]$^+$ calcd 309.2; found 309.2.

**GS-420 (isolated as the *bis*-HCl salt).** $^1$H NMR (400 MHz, Methanol-$d_4$) δ 8.92 (d, $J$ = 0.7 Hz, 1H), 7.81 (d, $J$ = 0.7 Hz, 1H), 4.52 (s, 2H), 4.45 (q, $J$ = 7.1 Hz, 2H), 4.29 (s, 2H), 3.83 (t, $J$ = 6.0 Hz, 2H), 3.47–3.36 (m, 4H), 2.98 (s, 6H), 1.41 (t, $J$ = 7.1 Hz, 3H), 1.26 (t, $J$ = 7.1 Hz, 3H). LCMS-ESI$^+$ (*m/z*): [M+H]$^+$ calcd 415.1/417.1; found 415.1/417.1.

**GS-444 (isolated as the *bis*-HCl salt).** $^1$H NMR (400 MHz, Methanol-$d_4$) δ 8.91 (d, $J$ = 0.6 Hz, 1H), 7.82 (d, $J$ = 0.7 Hz, 1H), 4.51 (s, 2H), 4.29 (s, 2H), 3.83 (t, $J$ = 6.0 Hz, 2H), 3.40 (dt, $J$ = 13.2, 6.5 Hz, 4H), 2.98 (s, 6H), 1.25 (t, $J$ = 7.1 Hz, 3H). LCMS-ESI$^+$ (*m/z*): [M+H]$^+$ calcd 387.1/389.1; found 387.1/389.1.

## PHH plating and culture conditions

Cryopreserved primary human hepatocytes (PHH) isolated from multiple donors were purchased from Thermo Fisher Scientific (HMCPTS; Donors Hu8181, Hu8130, Hu4167, Hu1748, Hu4239), Lonza (HEP187; Donor Hu7272, HUM4167), Bioreclamation (Donors BCD, VUZ), or Corning (Donors Hu276, Hu349, BD195). After thawing, cells were recovered by centrifugation at 100$g$ through cryopreserved hepatocyte recovery medium (Thermo Fisher Scientific; CM7500) and plated in collagen coated 96-well plates (Thermo Fisher Scientific; CM1096) at a density of 65,000–70,000 live cells per well. Cells were plated in William's E medium (Thermo Fisher Scientific; A1217601) supplemented with 3.6% hepatocyte thawing and plating supplement (Thermo Fisher Scientific, A15563), 5% fetal bovine serum (Thermo Fisher Scientific; 16000–036), 1 μM dexamethasone (Thermo Fisher Scientific, A15563), and 0.2% Torpedo antibiotic mix (Bioreclamation; Z990008). Approximately 12–14 hours after plating, plating medium was removed, and cells were switched into maintenance medium: William's E medium supplemented with 4% hepatocyte maintenance supplement (Thermo Fisher Scientific; AI15564), 2% fetal bovine serum, 0.1 μM dexamethasone, 1.5% DMSO (Sigma-Aldrich, St. Louis, MO; D8418), and 0.2% Torpedo antibiotic mix.

## HBV viruses

HepAD38 cells express HBV genotype D (GTD) virions under the control of an inducible tetracycline promoter [42]. For GTD virion production, HepAD38 cells were grown in DMEM-F12 medium (Thermo Fisher Scientific; 11320033) supplemented with 10% FBS; 1% Penicillin-Streptomycin-Glutamine; 1% HEPES, and 1% non-essential amino acids (Thermo

Fisher Scientific). Supernatant containing virions was collected every 3–4 days and virions were precipitated with PEGit (Systems Biosciences, Palo Alto, CA; LV810A-1) overnight at 4˚C. After precipitation, supernatant was spun at 3000 rpm at 4˚C for 15 minutes. The pellet containing the virions was resuspended in William's E medium containing 25% FBS. Viral titers were determined by measuring viral DNA by qPCR.

For additional HBV genotypes, sera (041FY67821P) from an HBV genotype A (GTA) infected patient and sera (024KY12630) from an HBV genotype E (GTE) infected patient were purchased from Proteogenex. Sera (56662-27867-39729-20130905) from an HBV genotype C (GTC) infected patient was purchased from BioCollections Worldwide, Inc. For the humanized mouse infection model, a GTC virus strain was used that was provided by PhoenixBio (Code No.: PBB004, Lot: 160205).

## PHH infection with HBV

Approximately 24 hours after plating, PHH were infected with HepAD38-derived GTD virus at 500 viral genome equivalents per cell in maintenance medium supplemented with 4% PEG 8000 (Promega, Madison, WI; V3011). For patient sera infections, PHH were infected with 6 μl of patient sera in maintenance medium supplemented with 4% PEG 8000. Infections were allowed to proceed for 20–24 hours before removing remaining extracellular virions by washing with maintenance medium three times.

## PHH compound treatment

Three days after infection with HBV virus (Day = 0), maintenance medium was replenished, and PHH were dosed in either the continuous or pulse dose regimen with 0, 0.016, 0.037, 0.080, 0.11, 0.33, 0.40, 1.0, 2.0, or 10 μM GS-5801 supplied in 100% DMSO using the HP Digital Dispenser D300 (Hewlett Packard, Palo Alto, CA). For the continuous dose experiment, cells received one dose of GS-5801 on days 0, 3, 6, and 10 and medium was not replenished until the next dose day. For the pulse dose regimen, medium containing compound was removed after two hours incubation and replaced with fresh maintenance medium every three to four days. Cells in the pulse dose experiment received only one dose of compound (Day = 0).

## Quantitation of extracellular HBV DNA in PHH assays

Viral DNA from PHH supernatants was purified using the Qiagen DNeasy 96 kit (Germantown, MD; 69582) following the manufacturer's recommended protocol. Briefly, 50 μl of supernatant was collected, lysed with an equal volume of Qiagen ATL buffer containing 10% proteinase K, and then incubated at 56˚C for 10 minutes. Lysates were then mixed with 100 μl of AE buffer containing ethanol, transferred into the DNeasy 96 well plate, and placed onto a vacuum manifold (Qiagen; 19504). The vacuum was applied to bind DNA to the DNeasy membrane while contaminants passed through. DNA bound to the membrane was washed first with 500 μl of AW1followed by 500 μl of AW2 buffer. After washing, plates were centrifuged (Sigma Model 4-16S) at 6000 rpm for 2 minutes. DNA was then eluted with 100 μl of pre-warmed AE buffer by a final centrifugation step of 6000 rpm for 2 minutes.

Quantification of vDNA by qPCR (quantitative polymerase chain reaction) amplification of the HBVX region of the genome was performed by combining 5 μl of DNA, 900 nM of forward primer (5'GGA CCC CTG CTC GTG TTA CA 3'), 900 nM of reverse primer (5'GAG AGA AGT CCA CCA CGA GTC TAG A–3'), 0.2 μM TaqMan probe (5' [6FAM] TGT TGA CAA GAA TCC TCA CCA ATA CCA C [NFQ-MGB] 3'), and 1X TaqMan Fast Advanced Master Mix (Thermo Fisher Scientific; 4444557) for a total reaction volume of 20 μl in 96-well PCR plates (Thermo Fisher Scientific; 4346906). qPCR was carried out on a real-

time PCR system (Thermo Fisher Scientific; QuantiStudio 7 Flex) using the following conditions: 95˚C for 20 seconds, followed by 40 cycles of 95˚C for 1 second and 60˚C for 20 second. A plasmid containing the HBV full genome was used for the standard curve.

## Quantification of intracellular HBV RNA in PHH assays

Intracellular HBV viral RNA was isolated from PHH using the RNeasy 96 kit (Qiagen, 74182) following the manufacturer's recommended protocol. Briefly, 125 μl of Qiagen RLT lysis buffer was added to PHH. The PHH lysate was then thoroughly mixed with 125 μl of 70% ethanol, transferred into a RNeasy 96 well plate, and placed onto a vacuum manifold (Qiagen; 19504). The plate was washed using RW1 and RPE buffers, followed by centrifugation (Sigma-Aldrich, Model 4-16S) at 6000 rpm for 2 minutes. The RNA was then eluted twice with 60 μl of nuclease free water for a total of 120 μl RNA.

After elution, DNase digestion by Turbo DNase (Thermo Fisher Scientific; AM2239) was performed to remove any contaminating DNA. After 30 minutes of DNase treatment at 37˚C, Turbo DNase was inactivated by adding 15 mM of EDTA and heating the reaction to 75˚C for 10 minutes in a thermo cycler (Thermo Fisher Scientific; Veriti 96 Well Thermal Cycler).

Quantification of vRNA by qRT-PCR (quantitative reverse transcription polymerase chain reaction) amplification of the HBVX region of the genome was performed by combining 5 μl of RNA to 900 nM of HBVX forward primer (5′ GGA CCC CTG CTC GTG TTA CA 3′), 900 nM of HBVX reverse primer (5′ GAG AGA AGT CCA CCA CGA GTC TAG A 3′), 0.2 μM TaqMan probe (5' [6FAM] TGT TGA CAA GAA TCC TCA CCA ATA CCA C [NFQ-MGB] 3'), and 1X glyceraldehyde 3-phosphate dehydrogenase (GAPDH) or ribosomal protein large P0 (RPLP0) endogenous transcripts (Thermo Fisher Scientific; 4310884E; 4310879E) and 1X TaqMan Fast Virus 1-Step Master Mix (Thermo Fisher Scientific; 4444434) for a total reaction volume of 20 μl in 96-well PCR plates (Thermo Fisher Scientific; 4346906). qRT-PCR was carried out on a real-time PCR system (Thermo Fisher Scientific; QuantiStudio 7 Flex) using the following conditions: 50˚C for 5 minutes, then 95˚C for 20 seconds, followed by 40 cycles of 95˚C for 3 second and 60˚C for 30 second.

*GAPDH* or *RPLP0* mRNA expression was used to normalize target gene expression. Levels of HBV RNA for all donors were calculated as fold change relative to no drug treated sample using the 2-ΔΔCt method [43].

## HBeAg and HBsAg quantification in PHH assays

Hepatitis B e antigen (HBeAg) and Hepatitis B surface antigen (HBsAg) were detected in culture media at the indicated time by ELISA or electrochemiluminescence assay (MSD). The HBeAg and HBsAg ELISAs were performed using the HBeAg ELISA kit (International Immuno-Diagnostics, Foster City, CA) and HBsAg ETI-MAK-2 plus kit (DiaSorin, Stillwater, MN), respectively according to the manufacturer's instructions. Concentrations were calculated by interpolation from standard curves with purified HBeAg and HBsAg. The MSD assay was performed according to the manufacturer's instructions (Meso Scale Diagnostics, Rockville, MD). Briefly, cultured supernatants were inactivated with 0.5% Triton X-100 (30 minutes at 37˚C) and then transferred into plates pre-spotted with both an anti-HBeAg antibody (Genway Bio, San Diego, CA) and a custom anti-HBsAg antibody. The plates were then incubated for two hours at room temperature with gentle shaking, followed by a wash step in PBS with 0.5% Tween. MSD sulfate tags anti-A and anti-B (1 μg/mL each) were then added to the wells and the plates incubated for a further two hours at room temperature with gentle shaking, followed by another wash step in PBS with 0.5% Tween. A 2X solution of MSD T Buffer Read was then added and the plate was read on a Sector Imager 6000 plate scanner.

## EC$_{50}$ determination

Antiviral activity of test compounds was determined from vRNA, vDNA, HBeAg, and HBsAg data by comparing compound-treated PHH to DMSO (vehicle)-treated PHH to generate a percent of DMSO control value (% DMSO control). The % DMSO control was calculated by the following equation:

$$\% \text{ DMSO Control} = 100 \times (X_c/X_D)$$

where Xc is the signal from the compound-treated PHH and X$_D$ is the signal from the DMSO-treated PHH. The % DMSO control for vRNA, vDNA, HBeAg, HBsAg, *GAPDH*, or *RPLP0* was plotted versus the log of each compound concentration in GraphPad Prism (version 6; Graphpad Software, LaJolla, CA) to generate dose-response curves. EC$_{50}$ values were defined as the test compound concentration that caused a 50% decrease in vRNA, vDNA, HBeAg, or HBsAg. CC$_{50}$ values were defined as the test compound concentration that caused a 50% decrease in *GAPDH* or *RPLP0*. Dose-response curves were fitted using the nonlinear regression equation "log(agonist) versus response–variable slope (four parameters)" in GraphPad Prism to determine EC$_{50}$ values.

## *KDM4/KDM5* siRNA knockdown

PHH from donor HUM4167 were plated in 96-well collagen coated plates at a cell density of 65,000 live cells per well as described above and PHH were infected with HepAD38-derived GTD HBV virions at 500 GE per cell. Three days after infection (Day 0) and nine days after infection (Day 6), PHH were transfected with siRNAs targeting *KDM5* or *KDM4* transcripts using Lipofectamine RNAiMAX (Thermo Fisher; 13778150) at a ratio of 6 pmol of siRNA to 1 μl RNAiMAX in OptiMEM media (Life Technology; 31985070). 30 μl of siRNA/RNAiMAX complexes were added to appropriate wells containing 150 μl of maintenance medium such that the final concentration of each individual siRNA was 2.5 nM. Three to four siRNAs were used per *KDM5* or *KDM4* gene to insure knockdown of all isoforms. A pool of 14 siRNAs was used to target transcripts of the *KDM5* family (Thermo Fisher 4392420; siRNA: s11834, s11835, s11836, s21144, s21145, s21146, s15748, s15749, s15750, s15775, s15776, s224895, s224896 and a custom *KDM5B* siRNA with sense sequence of 5'- `ACUUAUUCCUGUCCGGA-GAtt` -3' and anti-sense sequence of 5'- `UCUCCGGACAGGAAUAAGUtg` -3'). A pool of 16 siRNAs was used to target the transcripts of the *KDM4* family (Thermo Fisher 4392420; siRNA: s18635, s18636, s18637, s229325, s229326, s22867, s225929, s229931, s225930, s22990, s31266, s31267, s31268, s52751, s52752, s52753). Media was replenished with PHH maintenance media on days 3, 6, 7, and 10 post-initiation of siRNA transfection. On Day 0, 3, 6, 10, and 13 post-initiation of siRNA transfection, amounts of secreted HBeAg and HBsAg were measured to assess antiviral activity as described. In addition, on Day 13 alamarBlue (Thermo Fisher; DAL1100) was used as per the manufacturer's protocol to assess toxicity. Intracellular HBV RNA and *KDM* transcripts levels were measured by qRT-PCR to assess antiviral activity and siRNA knockdown. *KDM* primer/probe sets used for qRT-PCR were from Thermo Fisher (4392420): Hs00231908_m1, Hs00981910_m1, Hs01011846_m1, Hs00190491_m1, Hs00206360_m1, Hs00392119_m1, Hs00323906_m1, Hs00250616_s1, Hs00988859_s1, and Hs01096550_m1.

## cccDNA enrichment and Southern blotting

PHH were plated in 24-well plates at 350,000–400,000 live cells per well and infected with HBV genotype D as described above. PHH were treated with GS-5801 every three to four days

for 14 days prior to DNA isolation. DNA was isolated from PHH using a MasterPure Complete DNA Purification kit (Epicentre, Madison, WI; MC85200) according to the manufacturer's instructions, but omitting ProteinaseK or RNaseA treatment. After isolation, DNA was treated with T5 exonuclease (New England Biolabs, Ipswich, MA; M0363S) according to the manufacturer's instructions.

cccDNA and host mitochondrial DNA (ND4) levels were examined by Southern blot using branched DNA signal amplification (bDNA) method as previously described [44]. All probes and reagents were purchased from Thermo Fisher Scientific.

## Rat and cynomolgus monkey in vivo studies

**Liver to plasma GS-5801 and GS-080 AUC ratios.** The in vivo portion of the study for determination of liver to plasma drug loading ratios was performed in male Sprague Dawley rats and male cynomolgus monkeys at Covance (Madison, WI). Rats and monkeys were dosed by oral gavage once with 1 mg/kg or 2.5 mg/kg, respectively. The rat study included a vehicle control group (water). Serial venous blood samples were taken at 0, 0.25, 0.5, 1, 2, 4, 6, 8, 12, and 24 hours post dose from each animal and collected into vacutainer tubes containing potassium oxalate/sodium fluoride (Thermo Fisher; BD367925) as the anti-coagulant and blood was centrifuged for plasma isolation. For rat, livers were perfused with heparinized saline sodium nitrite solution immediately prior to harvest and liver samples were excised from vehicle and GS-5801 treated rats at sacrifice at 1, 8, and 24 hours post dose (n = 3 per time point). For monkey, liver samples were taken by biopsy at pre-dose, 1, 4, 24, 48, and 72 hours post dose (n = 2 per time point). Rat and monkey livers were weighed and immediately flash frozen in liquid nitrogen.

500 μL of cold HPLC grade water was added to each monkey liver biopsy sample and rat liver samples were diluted 3-fold with cold HPLC grade water. Diluted liver samples were homogenized between 1 and 3 minutes. 50 μL of plasma samples and homogenized liver samples were quenched with 200 μL of 0.05% formic acid in acetonitrile. Samples were then vortexed and centrifuged at 1500–3000 RPM for 15 minutes at RT and diluted as needed with HPLC grade water prior to quantification of GS-5801 and GS-080 by LC-MS/MS. An Agilent 1200 series binary pump (Santa Clara, CA; G1312A) was used for elution and separation of compounds with a Hypersil Gold C18 HPLC column (Thermo Fisher; 50 x 3.0 mm, 5 μm) over 6.75 minutes with 99–3% gradient of Mobile Phase A (1% acetonitrile in 2.5 mM ammonium formate aqueous solution pH 2.6) and 1–97% gradient Mobile Phase B (90% acetonitrile in aqueous 10 mM ammonium formate pH 6.8). GS-5801 and GS-080 were detected with a TSQ Quantum Ultra triple quadrupole mass spectrometer in selective reaction monitoring operation mode.

**Rat and monkey tissue PD (H3K4me3:H3).** The in vivo portion of the study for measurement of tissue PD (H3K4me3:H3 levels) was performed in male Sprague Dawley rats and male cynomolgus monkeys at Crown Bioscience (Taicang, China) and Covance (Madison, WI). Male Sprague-Dawley rats (n = 3 animals per dose group) were dosed p.o. with GS-5801 once daily for five days at 0.3, 1, 3, 10, or 30 mg/kg. Male cynomolgus monkeys (n = 2 or 3 animals per dose group) were dosed p.o. with GS-5801 once daily for seven days at 0.03, 0.1, 0.3, 1, 3, or 10 mg/kg. Animals were sacrificed 24 hours post final dose and approximately 100 mg of liver (left lateral lobe), lung, and kidney tissue was collected. Additionally, one mL of venous blood for PBMC isolation was collected into $K_2$EDTA vacutainer tubes (Thermo Fisher; BD367835). PBMCs were isolated from whole blood that was diluted with an equal volume of Dulbecco's Phosphate Buffered Saline without calcium or magnesium (DPBS; Lonza; 17-512F). Diluted whole blood was layered onto an equal volume of HISTOPAQUE-1077 (Sigma;

10771) gradient medium and centrifuged at 400 x g for 30 minutes with the centrifuge brake switch off. PBMCs were collected at the plasma-HISTOPAQUE-1077 interface and washed three times with DPBS by centrifugation at 500 x g for 10 minutes. Tissue samples and PBMCs were snap frozen using liquid nitrogen and stored at -80˚C for subsequent H3K4me3:H3 analysis by ELISA.

**Rat liver RNAseq study.**   The in vivo portion of the study for measurement of rat liver transcript levels by RNAseq was performed at Covance (Madison, WI). Male Wistar Han rats (n = 3 per dose group) were dosed p.o. once daily for seven days with GS-5801 at 10, 30, or 100 mg/kg and sacrificed 24 hours after the last dose. A second group of male Wistar Han rats (n = 3 per dose group) were dosed p.o. once daily for seven days with GS-5801 at 10, 30, or 100 mg/kg and taken off drug for seven days before animals were sacrificed. At sacrifice, two approximately 100 mg samples of liver tissue (left lateral lobe) were collected, snap frozen using liquid nitrogen, and stored at -80˚C for subsequent H3K4me3:H3 analysis by ELISA and transcriptome analysis by RNAseq.

## Care and use for cynomolgus monkeys at crown Biosciences, Taicang, China

**Housing conditions.**   The enrolled monkeys were housed and maintained in accordance with the guidelines approved by the Association for Assessment and Accreditation of Laboratory Animal Care (AAALAC). The targeted conditions for animal living environment and photoperiod were as follows: Temperature: 23 ± 3˚C Humidity: 50 ± 20% Light cycle: 12 hours light on and 12 hours light off.

**Diet and enrichment.**   All animals had free access to water and were fed twice daily with a complete, nutritionally balanced diet (Beijing Keao Xieli Feed Co., LTD, Beijing, China) enriched with seasonal fruits or vegetables.

**Steps to Alleviate suffering and clinical observations.**   Non-human primate care and use were conducted in accordance with all applicable assessment and accreditation of laboratory animal care (AAALAC) regulations and guidelines. Crown bioscience institutional animal care and use committee (IACUC) approved all animal procedures used in this study. All the procedures related to handling, care and treatment of the animals in this study were performed according to the guidelines approved by the association for AAALAC. After each treatment (weighing, bleeding or dosing), the animals were observed until the animals were able to stand up and alert if they were anesthetized. At the time of routine monitoring, the animals were checked for any effects of the compound on their behaviors such as mobility, body weight gain/loss, and any other abnormal activities. Clinical abnormalities observed and animal death were recorded and reported timely to the veterinarian, study director, and sponsor. Animals were closely monitored for any abnormal behavior, particularly vomiting, yawning, gaping, and signs of malaise or discomfort.

**Anesthesia and euthanasia.**   Animals were humanely euthanized at the end of the study. Those animals were injected with ketamine (15 mg/kg) plus xylazine (2 mg/kg) intramuscularly for anesthesia and operation at the times of being sacrificed for tissue collection.

## Care and use for cynomolgus monkeys at covance laboratories, Madison, WI

**Housing conditions.**   Animals were housed in stainless steel cages. When possible, animals were socially housed by sex: up to three animals/cage. Animals were individually housed during acclimation and for study-related procedures. Environmental controls for the animal room were set to maintain 20 to 26˚C, a relative humidity of 30 to 70%, a minimum of 10 air

changes/hour, and a 12-hour light/12-hour dark cycle. The light/dark cycle was interrupted for study-related activities.

**Diet.** Certified Primate Diet #2055C (Envigo RMS, Inc.) was provided one or two times daily unless otherwise specified. The diet is routinely analyzed by the manufacturer for nutritional components and environmental contaminants. Results of specified nutrient and contaminant analyses are on file at Covance-Madison. Water was provided ad libitum. Water samples are routinely analyzed for specified microorganisms and environmental contaminants. The results are on file at Covance-Madison.

**Environmental enrichment.** Animals were given various cage-enrichment devices; fruit, vegetable, or dietary enrichment (that do not require analyses). Animals were commingled as a form of environmental enrichment.

**Steps to alleviate suffering.** All study procedures were in accordance with the Animal Welfare Act, the Guide for the Care and Use of Laboratory Animals, and the Office of Laboratory Animal Welfare. Any medical treatment necessary to prevent unacceptable pain and suffering, including euthanasia, was the sole responsibility of the attending laboratory animal veterinarian. Palliative and prophylactic procedures were not needed on this study, as no abnormal observations were noted during health monitoring.

**Clinical observations.** Cage-side observations were performed once daily and detailed observations (removal of animal from cage) were performed weekly. Additionally, for health monitoring, animals were checked twice daily for mortality, abnormalities and signs of pain or distress; no abnormal findings were observed.

**Anesthesia and euthanasia.** Animals were humanely anesthetized at the end of the study with sodium pentobarbital and exsanguinated. Animals were sedated with ketamine for transport to the necropsy laboratory or for study-related procedures (ie: ECGs).

## Care and use for rats at covance laboratories, Madison, WI

**Housing conditions.** Animals were group-housed (up to five animals/cage) in polycarbonate cages with hardwood chip bedding. Animals were individually house in stainless steel or polycarbonate cages for study-related procedures.

**Diet.** Animals were offered Certified Rodent Diet #2014C (Envigo RMS, Inc.) ad libitum unless fasted for study procedures. Water was provided ad libitum.

**Environmental enrichment.** Animals were given various cage-enrichment devices and dietary enrichment.

## HBV mouse efficacy model

The in vivo and in vitro hepatocyte infection model portions of the study were performed by PhoenixBio (Higashi-Hiroshima, Japan). Mice used in this efficacy model (cDNA-uPAwild/+/SCID [cDNA-uPAwild/+: B6;129SvEv-Plau, SCID: C.B-17/Icr-*scid* /*scid* Jcl]) had an estimated human hepatocyte replacement index of 70% or more (PHH# BD195, Corning), which was calculated based on the blood concentration of human albumin prior to the inoculation as previously described [45]. Mice were infected with HBV GTC virus (PhoenixBio; PBB004, Lot #160205) and had serum HBV titer levels greater than 1.0E+08 copies/mL seven days prior to study initiation. HBV-infected mice were treated once daily (p.o.) for seven days (Day 0–6) with deionized water for acclimation. Mice (n = 8 per treatment group) received vehicle (deionized water), 30 mg/kg, or 100 mg/kg of GS-5801 p.o. once daily on Days 7 to 13, Days 21 to 27, Days 35 to 41, and Days 49 to 55. Fifty μL of blood was collected from animals under isoflurane anesthesia via the retro-orbital plexus/sinus and centrifuged for serum collection for HBsAg and HBV DNA measurements described below. At termination of study (Day 56),

100–200 mg of tissue was harvested from the left lateral liver lobe of all animals. Liver tissue samples were snap frozen using liquid nitrogen and stored at -80˚C for subsequent H3K4me3: H3 analysis by ELISA.

**HBsAg measurements from in vivo study.**  For in vivo mouse studies, serum HBsAg concentrations were determined by SRL, Inc. (Tokyo, Japan) based on the Chemiluminescent Enzyme Immuno Assay (CLEIA) developed by Fujirebio (Lumipulse® Presto II) [46].

**HBV DNA measurements from in vivo study.**  HBV DNA was extracted from 5 μL of serum using the SMITEST EX-R&D Nucleic Acid Extraction Kit (Medical & Biological Laboratories, CO. LTD, Nagoya, Japan) and dissolved in 20 μL nuclease-free water (Thermo Fisher Scientific Inc., Waltham, MA). Real-time PCR was used to measure the serum HBV DNA concentration using the TaqMan Fast Advanced Master Mix (Thermo Fisher Scientific) and ABI Prism 7500 sequence detector system (Applied Biosystems). The PCR reaction mixture was added into 5 μL of the extracted DNA. The initial activation of uracil-N-glycosylase at 50˚C for 2 minutes was followed by the polymerase activation at 95˚C for 20 seconds. Subsequent PCR amplification consisted of 53 cycles of denaturation at 95˚C for 3 seconds and annealing and extension at 60˚C for 32 seconds per cycle in an ABI 7500 sequence detector. The average serum HBV DNA level was calculated from the values of the two separate wells. Forward primer: CACATCAGGATTCCTAGGACC, Reverse primer: AGGTTGGTGAGTGATTGGAG, Probe: CAGAGTCTAGACTCGTGGTGGACTTC.

**Assessment of GS-5801 antiviral activity in PXB hepatocytes in vitro.**  Hepatocytes were isolated by two-step collagenase perfusion from PXB mice as described [47]. Hepatocytes were resuspended in DMEM medium (Thermo Fisher Scientific) containing 2% FBS (Biosera, Kansas City, MO), 20 mM HEPES (Thermo Fisher Scientific), 44 mN NaHCO$_3$ (Wako Chemicals, Richmond, VA), 100 IU/mL penicillin (Thermo Fisher Scientific), and 100 ug/mL streptomycin (Thermo Fisher Scientific) and plated at a density of 4 x 10$^5$ cells/well of a BioCoat Collagen I 24 well plate (Corning). Hepatocytes were infected with HBV virus GTC (PXB Strain PBB004, Lot; 20151109) at 5 genome equivalents per cell in DMEM inoculum medium containing 2% DMSO (Sigma-Aldrich), 4% PEG-8000 (Promega), 2% FBS (Biosera), 20 mM HEPES (Thermo Fisher Scientific), 44 mN NaHCO$_3$ (Wako Chemicals), 100 IU/mL penicillin (Thermo Fisher Scientific), and 100 ug/mL streptomycin (Thermo Fisher Scientific), 15 μg/mL L-proline (Wako Chemicals), 0.25 μg/mL insulin (Sigma-Aldrich), 50 nM dexamethazone (Sigma-Aldrich), 5 ng/mL epidermal growth factor (Sigma-Aldrich), and 0.1 mM L-ascorbic acid 2-phosphate (Wako Chemicals). After 24 hours, infection inoculum was removed, cells were washed once with DMEM + 2% FBS and medium was replaced with inoculum medium without PEG. Three days after infection, cells were treated with vehicle (DMSO) or 5-fold serial dilutions of GS-5801 (10–0.016 μM). Medium containing fresh compound was replaced every 3 days for a total of six doses of compound and levels of extracellular HBsAg and HBeAg were measured on Day 21.

**HBsAg and HBeAg measurements from in vitro hepatocyte study.**  Quantification of HBsAg and HBeAg levels was performed by SRL Inc. (Tokyo, Japan) based on the Chemiluminescent immunoassay (CLIA) using an ARCHITECT instrument and the ARCHITECT HBsAgQT (Abbott, Japan) and ARCHITECT HBeAgQT (Abbott, Japan) reagents [48].

## Care and use for mice at PhoenixBio, Higashi-Hiroshima, Japan

**Housing conditions.**  As One Corporation (Osaka, Japan) TP-107 cages with PaperClean (Shizuoka, Japan) sterilized soft paper were used to house one animal per cage. Cages were sanitized once weekly. The targeted conditions for animal living environment and photoperiod were as follows: Temperature: 23 ± 5˚C Humidity: 55 ± 25% Light cycle: 12 hours light on and 12 hours light off.

**Diet.** Sterilized CRF1 containing Vitamin C pellet diet (Oriental Yeast Co., Ltd, Tokyo, Japan) was provided ad libitum. Sterilized water was provided ad libitum.

**Steps to alleviate suffering and clinical observations.** The use of animals on this study and experimental procedures were approved by the Animal Ethics Committee of PhoenixBio. Detailed observations of health condition and body weight measurements were conducted daily.

**Anesthesia and euthanasia.** Animals were anesthetized with isoflurane anesthesia and sacrificed by exsanguination.

## H3K4me3:H3 ELISA

Histone extractions and H3K4me3 and H3 measurements from rat and cynomolgus monkey tissue and PBMCs were performed at Crown Bioscience (Taicang, China).

**Histone enrichment.** Histones were enriched from tissue and PBMC samples using the AbCam total histone extraction kit (Cambridge, UK; ab113476) according to the manufacturer's protocol after cell/tissue dissociation with an IKA T10 TURRAX homogenizer (Wilmington, NC; EW-04720-50). Histones were enriched from 500,000 to 750,000 PHH. PHH monolayers were washed once with cold (4˚C) 1X DPBS (Dulbecco's phosphate buffered saline; 21-031-CV, Corning Inc.). Cold triton extraction buffer (TEB): 1X DPBS, 0.5% triton X-100 (Sigma-Aldrich, T9284), 2 mM phenylmethylsulfonyl fluoride (PMSF; Thermo Fisher Scientific, 36978), was added to the cell monolayer at a density of $1 \times 10^6$ cells/mL. After 10-minute incubation in TEB at 4˚C, the cell monolayer was scraped and cells were pelleted by centrifuged at 1200 rpm at 4˚C for 10 minutes. The cell pellets were washed again in TEB by centrifugation at 1200 rpm at 4˚C for 10 minutes. The resulting cell pellet was re-suspended in cold (4˚C) 0.2 Normal hydrochloric acid (HCl; Sigma-Aldrich, 343102) at a density of ~ $1 \times 10^7$ cells/mL and incubated at 4˚C for 30 minutes. After incubation, HCl extractions were pelleted by centrifugation at 1200 rpm for 5 minutes at 4˚C and supernatant containing the enriched histones was collected.

Total protein concentration of all samples following histone enrichment was determined using a bicinchoninic acid (BCA) assay (Thermo Fisher Scientific; 23225).

**H3K4me3:H3 ELISA.** An indirect ELISA assay was used to measure amounts of H3K4me3 relative to total H3 (H3K4me3:H3) extracted from tissue, PBMC, or PHH samples. Amounts of H3, H3K4me3, and non-specific background from the secondary antibody (blank absorbance) were measured for each sample in independent ELISA wells. Briefly, each histone extraction from tissue or PBMC lysates was diluted in 50 mM $Na_2CO_3$ (Sigma-Aldrich; 223484), pH 9.5 coating buffer and 50 µl of each diluted sample was added to 384-well plates (Greiner, Kremsmünster, Austria; 781061) in duplicate. A coating concentration for tissue and PBMC histone extracts was used such that the H3 and H3K4me3 OD measurements were in the linear range for the H3 and H3K4me3 antibodies. After plates were incubated overnight at 4˚C to allow for coating of histone samples, plates were blocked for one hour at room temperature (RT) in 100 µl 5% skimmed milk (Fluka; 70166)/1X DPBS (Gibco; 70011–044) % w/v solution. Plates were washed three times with 0.1% Tween-20 (Amresco; 0777)/1 X DPBS % v/v solution (DPBST) and 50 µl/well of 0.05 µg/ml H3 (Sino Biological; 11231-RP02) or 0.5 µg/ml H3K4me3 (Millipore; 07473) primary antibodies diluted in 1X DPBS were added to the ELISA plates and incubated for one hour at RT. After primary antibody incubation, plates were washed 3 times with 100 µl/well PBST and 50 µl/well of an AbCam secondary antibody (ab137914) diluted to 0.05 µg/ml in 1X PBS was added to the ELISA plates and incubated for one hour at RT. Plates were washed five times with 100 µl/well DPBST and 50 µl/well 3,3′,5,5′-tetramethylbenzidine (TMB) substrate solution (Sigma; T0440) was added to each plate. Upon

color development (~ three–five minutes), reactions were quenched by adding 50 μl/well of 1 M $H_2SO_4$ (Sigma; 72266). Absorbance of each well was measured at 450 nm within five minutes using a SpectraMax Plus 384 Microplate Reader (Molecular Devices; PLUS 384).

### Histone mass spectrometry

**Histone purification from PHH.** Approximately 25 million PHH from Donor BCD (Bioreclamation) were plated on collagen coated (Coating Matrix kit, Life Technologies, Cat. #R-011-K) plates (coated for 30 minutes at 37˚C and subsequently washed twice with PHH plating medium). The PHH plating medium was removed after 16 hours and replaced with PHH maintenance medium (Day 0). Fresh maintenance medium supplemented with DMSO or 10 μM GS-5801 was added 3 days later (Day 3). The maintenance medium was refreshed again with 10 μM GS-5801 or DMSO control on Day 6, and the histones were extracted and purified 24 hours later (Day 7) using the Active Motif histone purification mini kit (Cat. #40026). Modifications to the manufacturer's protocol included two PBS washes prior to adding the histone extraction buffer to the cells, precipitation of eluted histones in 8% perchloric acid, and a single wash in cold acetone containing 0.2% HCl.

**Propionic anhydride labeling.** 3 μg of purified core histones from control and GS-5801 treated PHH were diluted with deionized $H_2O$ to a total volume of 9 μl and buffered to pH 8.5 by addition of 1 μl of 1 M triethylammonium bicarbonate buffer. Fresh 1:100 propionic anhydride:water reagent was prepared and 1 μl of mixture was added immediately to the histone sample with vortexing and incubated for 5 minutes at 25˚C. The reaction was quenched with 1 μl of 80 mM hydroxylamine for 20 minutes at 25˚C. Tryptic digestion was performed overnight with 0.3 μg trypsin (Promega Sequencing Grade; Madison, WI) per sample. Samples were dried down in a speed vac and resuspended in 20 μl 0.1 M TEAB. To label the peptide N-termini, fresh 1:100 propionic anhydride (light) and 1:100 d10-propionic anhydride (heavy) in water were prepared and used to label the control or GS-5801 treated digests by adding 2 μl and incubating samples for 5 minutes at 25˚C. The reaction was quenched with 1 μl of 80 mM hydroxylamine for 20 minutes at 25˚C, dried down in a speed vac, and the resulting control and treated samples were resuspended and combined in 50 μl of 3% ACN and 0.1% formic acid for analysis by mass spectrometry.

**Mass spectrometry.** 1 μl of sample was injected using a Thermo Fischer Scientific Ultimate 3000 Autosampler (San Jose, CA) onto a 15 cm x 75 μm Thermo Fischer Scientific ES812 Easyspray nano column with 5 μm particle size. Separation was performed using an Thermo Fischer Scientific Ultimate nano 3000 UHPLC with a 300 nl/min flow rate (solvent A (0.1% formic acid) and solvent B (90% acetonitrile, 0.1% formic acid)) with a gradient from 3% solvent B to 35% (60 min), then 3% to 9% (15 min), holding at 90% for 5 min, and finally re-equilibrating for 20 min at 3% solvent B. Mass spectrometric analysis was performed using a ThermoFischer Scientific Q-Exactive HF using a data dependent acquisition. Precursor scans from 350–1200 m/z as performed at 120 K resolution using 50 ms maximum fill time and 3e6 AGC target. The top 15 ions were selected for MSMS analysis at 15 K resolution using normalized collision energy of 27 with a 50 ms fill time and 1E5 AGC target. Each sample was analyzed three times.

**Data analysis.** Data was searched using Thermo Fischer Scientific Proteome Discoverer V2.1 software. Peak intensities corresponding to unmodified, methylated, dimethylated, and trimethylated H3K4 peptide "TKQTAR" were manually extracted using ThermoFischer Scientific Xcaliber software. The ratio of drug treated/control was calculated for each sample and then corrected using the average protein level ratio to account for differences in total protein between control and treated samples.

## Biochemical characterization

**Biochemical reagents.** S3 Table in S1 File contains detailed information about the sources of the recombinant enzymes. Peptide substrates were from AnaSpec (Fremont, CA). α-Ketoglutarate (disodium salt dihydrate, 75892), L-ascorbic acid (A0278), ammonium iron (II) sulfate hexahydrate (215406), BSA (A3803), and β-nicotinamide adenine dinucleotide (NAD, N6522) were from Sigma-Aldrich. Tween-20 (10%, 28320) was from Thermo Fisher Scientific. 384 Alpha-plates, acceptor beads (5 mg/mL), alpha streptavidin donor beads (5 mg/mL, 6760002), and 5× Epigenetic buffer (AL008) were from PerkinElmer (Waltham, MA). Chicken core histones were obtained from Millipore (Billerica, MA). HeLa cell nucleosomes were from Reaction Biology Corporation (Malvern, PA). Nucleosome antigen was from Arotec Diagnosis (ATN02, Wellington, New Zealand). Small molecule inhibitors used as positive controls included S2101 (CalBiochem/Millipore; 489477), 2,4-Pyridinedicarboxylic acid (AK Scientific, Union City, CA; 00473), 8-hydroxyquinoline (Sigma-Aldrich; S018), 8-hydroxy-5-quinoline-carboxylic acid (Sigma-Aldrich; SML0057), S-(5′-Adenosyl)-L-homocysteine (Sigma-Aldrich; A9384), Trichostatin A (Sigma-Aldrich; T8552), and EX-527 (Sigma-Aldrich; E7024).

**KDM enzymatic assays.** The following assay protocol was used for all histone demethylases tested except KDM1A, KDM5A, B, and D, which are described below. Compounds were serially diluted into a 384-well Alpha plate as a DMSO solution and mixed with 5 µL reaction buffer (buffer A) containing 50 mM HEPES (pH 7.0), 0.1% BSA, and 0.003% Tween-20. The enzyme of interest was pre-incubated with 15 µM of $Fe^{2+}$ for 10 min (except 30 min for KDM2B) in reaction buffer A. 5 µL of the enzyme mixture was added to each of the compound-containing wells and incubated for 10 min. The reaction was started by addition of a 5 µL mixture containing 300 nM peptide substrate, 75 µM L-ascorbic acid, and 30 µM α-ketoglutarate in buffer A. This final 15 µL of reaction mixture was incubated for 20 min for KDM5B and 1 hour for the other KDM enzymes. The reaction was quenched with 10 µL of diluted AlphaLISA Acceptor beads (1:400) in Epigenetics Buffer and incubated for 60 min. The Donor beads were diluted 1:400 into Epigenetics Buffer in the dark and 10 µL of this was added to each well and incubated for another 60 min in the dark. The assay plates were read on an Enspire plate reader (PerkinElmer) using a standard AlphaLISA program.

KDM5A and 5B were tested using CisBio HTRF assay platform by Reaction Biology Corp (Malvern, PA). All concentrations are final unless noted otherwise. Inhibitors were pre-incubated with 2.5 nM KDM5A or 1.2 nM KDM5B in a 5 µL reaction mixture containing 50 mM HEPES (pH 7.5), 50 mM NaCl, 0.01% Tween 20, 0.01% BSA for 15 min temperature. The reaction was initiated by addition of 5 µL of 50 nM Biotin-H3K4me3 and incubated with gentle shaking for 45 min at RT. The reaction was quenched with 10 mM EDTA and 200 mM potassium fluoride. Upon addition of Eu-Antibody and Streptavidine-XL665, the mixture was incubated for 2 hours in the dark. The fluorescence resonance energy transfer was measured using an excitation wavelength of 320 nm and emission wavelength of 665 nm on an Envision plate reader (Perkin Elmer).

KDM5D was tested using a LANCE TR-FRET assay by Eurofins Cerep SA (France). All concentrations were final unless noted otherwise. Inhibitors were added to a mixture containing 45 mM HEPES/Tris-HCl (pH 7.5), 5 µM ferrous ammonium sulfate, 100 µM ascorbic acid, 10 µM 2-oxoglutarate, 0.01% Tween 20, 0.01% BSA, and 10 ng of enzyme, followed by addition of 100 nM biotin-H3K4me3 substrate. The 10 µL reaction mixture was incubated for 10 min at room temperature and quenched through the addition of 1 mM EDTA to reach a final concentration of 0.33 mM EDTA. After 5 min, the Eu-labeled anti-methyl histone H3K4me1-2 antibody and the Ulight streptavidin were added, and the mixture was incubated for 60 min. The fluorescence resonance energy transfer was measured using an excitation wavelength of 320

nm and emission wavelengths of 620 nm and 665 nm on an Envision plate reader (Perkin Elmer).

The KDM1 assay was conducted in the presence of NAD but without Fe(II) or ascorbic acid. A 10 μL reaction mixture containing 50 mM Tris-HCl (pH 9.0), 50 mM NaCl, 0.01% Tween-20, 0.25 nM of enzyme, and 1 mM NAD was pre-incubated for 10 min. The reaction was started by addition of 5 μL of H3(1–21)K4 me1, and the reaction mixture was incubated for 60 min. The reaction was quenched with 10 μL AlphaLISA Acceptor beads (1:200) in Epigenetics Buffer and the reaction mixture was incubated for 60 min. The Donor beads were diluted 1:200 into Epigenetics Buffer in the dark, and 10 μL of this mixture was added to each well and incubated for another 60 min in the dark. The assay plates were read on an Enspire plate reader (PerkinElmer) using a standard AlphaLISA program.

**Histone methyltransferase enzymatic assays.** Histone methyltransferase assays were performed in 96-well half-area optiplates (PerkinElmer) at room temperature. All concentrations are final unless noted otherwise. Compounds were serially diluted in DMSO and the final DMSO concentration was 1%. A 24 μL reaction mixture contained compound, enzyme, 200 nM of nucleosome or histone substrate, 400 nM $^3$H-SAM (83.1 Ci/mmol), 50 mM Tris-HCl (pH 8.0), 50 mM NaCl, 5 mM $MgCl_2$, 1 mM DTT, and 0.01% Tween-20. The plate was sealed, mixed on a Titramax plate shaker at 1200 rpm for 30 sec, and incubated at room temperature for 60 min. The reaction was quenched by addition of 24 μL mixture containing 5 mg/mL PVT-PEI beads and 300 μM unlabeled SAM in water. The plate was sealed, mixed as previously noted, incubated overnight, and read on a TopCount plate reader (PerkinElmer).

**Histone deacetylase enzymatic assays.** The HDAC-1 histone deacetylase assay was tested using the FLUOR DE LYS platform (Cisbio, Bedford, MA). Inhibition of SIRT-1 histone deacetylase was tested with the AlphaLISA detection system. All assays were conducted at room temperature with a final DMSO concentration of 1%. The HDAC-1 reactions were started by incubating a 5-μL mixture containing 50 mM Tris-HCl (pH 8.0), 137 nM NaCl, 2.7 mM KCl, 1 mM $MgCl_2$, 1 mg/mL BSA, 10 nM HDAC-1, and various concentrations of inhibitors for 10 minutes prior to addition of 5 μL 100 μM fluorogenic peptide from p53 residues 379–382 [RHKK(Ac)AMC]. After a 60-120-minute incubation at 30 ºC, the reaction was quenched with 2 μM trichostatin A (final concentration), mixed with 10 μL of Developer and incubated at 30 ºC for 1 hr. The fluorescence signal was measured using an excitation wavelength of 360 nm and emission wavelength of 460 nm on an EnVision Multilabel plate reader (PerkinElmer). The SIRT-1 reactions were carried out in a 15 μL mixture containing 50 mM Tris-HCl (pH 8.0), 150 nM NaCl, 1 mM DTT, 0.01% Tween-20, 0.01% BSA, 5 nM enzyme, 1 mM NAD, and 25 nM H3(1–21)K4(Ac) substrate. After a 60-minute incubation, the reaction was quenched with 10 μL AlphaLISA H3K4-specific Acceptor beads (1:200) in Epigenetics Buffer, and the reaction mixture was incubated for 60 min. The Donor beads were diluted 1:200 into Epigenetics Buffer in the dark and 10 μL of this mixture was added to each well and incubated for another 60 min in the dark. The assay plates were read on an Enspire plate reader (PerkinElmer) using a standard AlphaLISA program.

**IC50 determination.** The IC50 value was defined as the concentration of inhibitor inducing 50% decrease in product formation. Data was analyzed using GraphPad Prism 6.0 (La Jolla, CA). Unless otherwise mentioned, IC50 values were calculated by non-linear regression analysis using sigmoidal dose-response (variable slope) equation (four parameter logistic equation):

$$Y = Bottom + (Top - Bottom)/(1 + 10^\wedge((LogIC50 - X) * HillSlope)),$$

where X is log of the concentration of the inhibitor, Y is the response, the Bottom and Top

values were fixed at 0% and 100%, respectively. IC50 values were calculated as an average of at least two independent experiments.

### RNA-Seq library preparation and analysis

Library building and sequencing for RNA-seq of PHH and rat liver was performed on all samples at EA Genomics/Q2 Solutions (Morrisville, NC). Approximately 100 mg of rat liver tissue from the left later lobe was flash frozen immediately after collection. PHH were lysed into RLT buffer (Qiagen; 79216) and immediately frozen. Liver/PHH samples were homogenized in QIAzol Lysis Reagent (79306) and after addition of chloroform the samples were separated into aqueous and organic phases by centrifugation. Ethanol was added to the upper aqueous phase and the sample was applied to an RNeasy Mini spin column (74104) for purification and RNA eluted with water. Libraries were prepared using the Illumina TruSeq stranded mRNA sample preparation kit according to the manufacturer's protocol (Illumina, Inc., Hayward, CA; RS-122-2103). Approximately 30 million 2 x 50 bp paired-end reads were collected per sample using the Illumina HiSeq2000 sequencing platform. Sequencing reads were aligned to the human, rat, or HBV genomes by STAR [49]. The Bioconductor packages edgeR [50] and limma [51] were used to normalize sequence count data (counts per million; cpm) and conduct differential gene expression analysis. The false discovery rate (FDR) was calculated using the Benjamini-Hochberg method [52].

## Supporting information

**S1 File.**
(DOCX)

**S1 Data.**
(XLSX)

## Acknowledgments

We thank Gary Lee and Hoa Truong for performing the MSD immunoassay; Lindsay Gamelin for PHH donor screening as well as assistance identifying patient serum samples, as well as Tomas Cihlar, Anuj Gaggar, and Vithika Suri for helpful suggestions and discussions.

## Author Contributions

**Conceptualization:** Sarah A. Gilmore, Chelsea Snyder, Julie Farand, Ryan Dick, Todd C. Appleby, Gabriel Birkus, Tetsuya Kobayashi, Marc Labelle, Thomas Boesen, William E. Delaney, IV, Gregory T. Notte, Uli Schmitz, Becket Feierbach.

**Data curation:** Sarah A. Gilmore, Danny Tam, Tara L. Cheung, Chelsea Snyder, Julie Farand, Ryan Dick, Mike Matles, Ricardo Ramirez, Dwight Barnes, Todd C. Appleby, Gabriel Birkus, Madeleine Willkom, Tetsuya Kobayashi, Chin H. Tay, Gregory T. Notte, Uli Schmitz, Becket Feierbach.

**Formal analysis:** Sarah A. Gilmore, Danny Tam, Tara L. Cheung, Chelsea Snyder, Julie Farand, Ryan Dick, Joy Y. Feng, Ricardo Ramirez, Helen Yu, Yili Xu, Dwight Barnes, Gregg Czerwieniec, Katherine M. Brendza, Todd C. Appleby, Gabriel Birkus, Madeleine Willkom, Tetsuya Kobayashi, Eric Paoli, Chin H. Tay, Gregory T. Notte, Uli Schmitz, Becket Feierbach.

**Investigation:** Sarah A. Gilmore, Danny Tam, Tara L. Cheung, Chelsea Snyder, Julie Farand, Ryan Dick, Mike Matles, Joy Y. Feng, Ricardo Ramirez, Li Li, Helen Yu, Gregg Czerwieniec, Katherine M. Brendza, Todd C. Appleby, Gabriel Birkus, Tetsuya Kobayashi, Eric Paoli, Chin H. Tay, Gregory T. Notte, Uli Schmitz, Becket Feierbach.

**Methodology:** Sarah A. Gilmore, Li Li, Gabriel Birkus, Tetsuya Kobayashi.

**Project administration:** Sarah A. Gilmore, Li Li, Todd C. Appleby, Gabriel Birkus, Gregory T. Notte, Uli Schmitz, Becket Feierbach.

**Resources:** Tetsuya Kobayashi.

**Supervision:** Chelsea Snyder, Gabriel Birkus, William E. Delaney, IV.

**Validation:** Gabriel Birkus, Tetsuya Kobayashi, Gregory T. Notte.

**Visualization:** Gregory T. Notte.

**Writing – original draft:** Sarah A. Gilmore, Becket Feierbach.

**Writing – review & editing:** Sarah A. Gilmore, Uli Schmitz, Becket Feierbach.

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
