## [Decision Letter · Decision Letter 0]

26 Aug 2022

PONE-D-22-15852Characterization of a KDM5 Small Molecule Inhibitor with Antiviral Activity against Hepatitis B VirusPLOS ONE

Dear Dr. Schmitz,

Thank you for submitting your manuscript to PLOS ONE. After careful consideration, we feel that it has merit but does not fully meet PLOS ONE’s publication criteria as it currently stands. Therefore, we invite you to submit a revised version of the manuscript that addresses the points raised during the review process.

We look forward to receiving your revised manuscript.

Kind regards,

Haitao Guo

Academic Editor

PLOS ONE

Journal Requirements:

2. Please ensure you have stated in the 'Care and Use for Rats at Covance Laboratories, Madison, WI' section of your manuscript text (from line 910) the method(s) of animal sacrifice used in that section of the study.

“Funding for this research was provided by Gilead Sciences, Inc. D.T., T.C, C.S., J.F., M.M., J.Y.F., R.R., L.L., H.Y., Y.X., D.B., T.A., U.S., G.N., and B.F. are employees of Gilead Sciences, Inc. S.G., T.K., M.W., R.D., G.C., K.B., G.B., E.P., C.T., W.D. were previously employed by Gilead Sciences, Inc. All authors may hold stock or stock options in Gilead Sciences, Inc.”

Reviewers' comments:

Reviewer's Responses to Questions

**Comments to the Author**

1. Is the manuscript technically sound, and do the data support the conclusions?

Reviewer #1: Partly

Reviewer #2: Yes

2. Has the statistical analysis been performed appropriately and rigorously? 

Reviewer #1: Yes

Reviewer #2: Yes

3. Have the authors made all data underlying the findings in their manuscript fully available?

Reviewer #1: Yes

Reviewer #2: Yes

4. Is the manuscript presented in an intelligible fashion and written in standard English?

Reviewer #1: Yes

Reviewer #2: Yes

5. Review Comments to the Author

Reviewer #1: In this article the authors identify a nicotinic acid derivative post-screening as a KDM5 inhibitor with anti-HBV activity. This prodrug, GS-5801, was found to selectively inhibit KDM5 activity leading to accumulation of tri-methylation on H3K4 which in turn negatively impacted HBV replication and HBV antigen production in primary human hepatocytes. However, they drug failed to achieve the antiviral effects in the humanized mice model as observed in the in vitro set up.

The authors are recommended to provide a more quantitative comparative characterization of the HBV replication (as a proof that the inhibitor does/doesn’t influence the replication) and cccDNA formation through northern blots for the total RNA and representation of the core DNA and Hirt DNA (to visualize the rc-DNA as well the corresponding cccDNA.

The KDM4-5 knockdown upon siRNA-treatment is underwhelmingly low which raises the question of off-target effects leading to an indirect reduction in HBV RNA and Antigen levels.

A comment on effect on the replication cannot be made solely based on HBV RNA qRT-PCR and HBV AG immunoassay. The authors are recommended to perform southern blots for coreDNA and hirtDNA

Conclusion made on page 10 lines 208-210 cannot be made because the authors are just validating the effect of inhibiting KDMs upon HBV transcription. The future figures in some way confirm the role of GS-5801 as an antiviral and then may the conclusion be made but not this early in the story.

Fig 5c: Will the authors shed more light on what they mean here: Page 13 lines 292-294. Was the HBeAg measure on Day 3 post pulse dosing? If not, how does Day 12 low H3K4me3:H3 ratio correlate to increase inhibition of HBeAg release?

Please include label for figure legend 5(C) on Page 14 line 311 and please provide a figure legend for Fig 5(D)

Comments on observations made in 5(B) where the H3K4me3 maintenance doesn’t prolong while the antiviral activity does? Have the authors looked at the half-life of the GS-5801? Is the trimethylation of H3K4 lost around Day 9 and if so, how?

Why did the authors choose to perform continuous dosing instead of pulse-dosing in the future experiments in Fig 6?

Fig 6: Can the authors please elaborate on what kinds of effect on the viral transcriptome upon GS-5801 treatment.

Recommendation for a better representation of the host transcriptome changes observed in Fig 6(D) through highlighting specific pathways and genes on the heat map.

Pathway changes may be shown as bar graphs as downregulated and upregulated

Reviewer #2: Gilmore et al report in this manuscript the identification of small molecular inhibitors of histone lysine demethylase 5 (KDM5), GS-080 and its ester prodrug GS-5801, significantly reduced the levels of intracellular HBV RNA, DNA and secreted HBsAg and HBeAg without alteration of cccDNA amount in HBV infected primary human hepatocyte (PHH) cultures. In vitro enzymatic profiling showed that GS-5801 selectively inhibited KDM5, with highest potency against KDM5A and 5B. In agreement with the pharmacological results, siRNA knockdown of the transcripts of each individual KDM5A, B, C or D modestly reduced viral RNA and antigen secretion. However, simultaneous knockdown of KDM5A, B, C and D transcripts, but not other KDMs, results in significant reduction of HBV RNA and antigen secretion in HBV infected PHHs. Those results strongly support the hypothesis that KDM5 play a critical role in HBV cccDNA transcription and pharmacological inhibition of KSM5 significantly inhibits cccDNA transcription in PHHs. However, evaluation of GS-5801 antiviral activity in HBV infected humanized mouse model repopulated with the same batch of PHHs did not demonstrate any detectable effects on viral gene products, despite increased ratio of H3K4me3:H3 in the humanized livers.

Overall, the study is well conceived and executed. The discrepancy between the in vitro and in vivo efficacy had been thoroughly discussed.

My only suggestion is that a comparative ChIP analysis of cccDNA-associated H3K4 methylation and other epigenetic markers in cultured PHHs and human hepatocytes in chimeric mice under mock and GS-5801 treatment may provide clues on the discrepancy of in vitro and in vivo antiviral efficacy of GS-5801. However, this work is not essential for the publication of this work.

6. PLOS authors have the option to publish the peer review history of their article (what does this mean?). If published, this will include your full peer review and any attached files.

Reviewer #1: No

Reviewer #2: No

---

## [Author Response · Author response to Decision Letter 0]

14 Nov 2022

aOct 6, 2022

PONE-D-22-15852

Characterization of a KDM5 Small Molecule Inhibitor with Antiviral Activity against Hepatitis B Virus

Dear Prof. Guo,

On behalf of my co-authors, I want to thank you and the reviewers for helping us with turning this complex, disappointing research experience into something that the scientific community might benefit from. 

The “non essential” ChIP experiment suggested by reviewer 2 would indeed be very interesting. We had considered it early on, but the plan was abandoned after it became clear that the compound had no in vivo antiviral activity. 

Below are our responses toward the points raised by reviewer 1:

The authors are recommended to provide a more quantitative comparative characterization of the HBV replication (as a proof that the inhibitor does/doesn’t influence the replication) and cccDNA formation through northern blots for the total RNA and representation of the core DNA and Hirt DNA (to visualize the rc-DNA as well the corresponding cccDNA.

A comment on effect on the replication cannot be made solely based on HBV RNA qRT-PCR and HBV AG immunoassay. The authors are recommended to perform southern blots for coreDNA and hirtDNA

This is indeed a good point, as we have used the terms “replication inhibitor” and “transcription inhibitor” almost interchangeably, even in the Abstract. But clearly, inhibitors of epigenetic modulators might play a role in how HBV genomic DNA becomes histone associated cccDNA, but have a much bigger chance to affect gene expression from cccDNA. Our goal was to find small molecules that could suppress viral gene expression and thereby limit the spread to uninfected cells and possibly allowing a differential immune response.

We have clarified this in the text throughout.

We also note that on page 6 the analysis of cccDNA levels is discussed referring to S1 Fig. in Suppl Material. This shows the southern blots for cccDNA and mtDNA after treatment with T5 exonuclease. This might not be the exact experiment that reviewer 1 was pointing, but it is commonly used. There is no sign of reduced cccDNA, suggesting that GS-5801 is not a clear inhibitor of the formation of cccDNA. It is not a true “replication inhibitor”.

The KDM4-5 knockdown upon siRNA-treatment is underwhelmingly low which raises the question of off-target effects leading to an indirect reduction in HBV RNA and Antigen levels.

Indeed, the knockdown levels are low. But this data must be taken in the context of the GS-5801 selectivity footprint in the KDM space. Of course, the compound could be hitting an unknown target that we have not tested for. However, seeing qualitatively similar results with the siRNA knockdown of the KDMs makes it much more likely that the KDMs’ have something to do with the antiviral activity of the compound (without proving it absolutely). The data together allowed us to move forward to plan the pharmacodynamic characterization using methyl mark changes. 

Conclusion made on page 10 lines 208-210 cannot be made because the authors are just validating the effect of inhibiting KDMs upon HBV transcription. The future figures in some way confirm the role of GS-5801 as an antiviral and then may the conclusion be made but not this early in the story.

We softened the language a bit. It is not presented as a conclusion.

To explain that statement on page 10, the antiviral activity of the compound as been described along with the in vitro profile of KDM inhibition before. We follow with the observation that siRNA knockdown of KDM5a-d results in a similar, delayed antiviral effect. Wouldn’t that suggest the following which is the original wording: “Together these data suggest that GS-5801 targets KDM5 to cause antiviral activity”

Please include label for figure legend 5(C) on Page 14 line 311 and please provide a figure legend for Fig 5(D)

Done (apologies, cut and paste error in original document)

Fig 5c: Will the authors shed more light on what they mean here: Page 13 lines 292-294. Was the HBeAg measure on Day 3 post pulse dosing? If not, how does Day 12 low H3K4me3:H3 ratio correlate to increase inhibition of HBeAg release?

Some of the lack of clarity might be related to the missing legend for half of Figure 5. But it was poorly worded and text has been changed.

In Fig 5 A and B, antigen and H3K4me3:H3 ratios come from the same sample. %inh for HBeAg was measured on day 3, after which the compound was washed out in Fig 5B. PHH exposure to GS-5801 for 3 days (= pulsed) hence leads to the same level of suppression of HBeAg on day 12 as dosing for all 12 days (continuous, which means replenishing compound after day 3, 6 and 9).

Comments on observations made in 5(B) where the H3K4me3 maintenance doesn’t prolong while the antiviral activity does? Have the authors looked at the half-life of the GS-5801? Is the trimethylation of H3K4 lost around Day 9 and if so, how?

The H3K4me3:H3 ratios is indeed almost back to baseline levels by day 9, which is six days without compound. We assume that this process is the result of the removal of the compound, such that KDM5 can take the system back to steady state.

In vitro stability was indeed investigated as hepatocytes typically metabolize compounds in minutes or hours. GS-5801 is an ester prodrug from which the very stable GS-080 is generated rapidly. The PHH half-life of GS-080 (1 uM) in uninfected PHH is 42-66 hrs. 

We added a paragraph to the ms pointing out what this means for clinical development.

Why did the authors choose to perform continuous dosing instead of pulse-dosing in the future experiments in Fig 6?

The true answer is that we ran those studies before it was clear that short pulse dosing is possible. We did not rerun the RNAseq experiment/analysis as day 3 would be expected to be similar to the data we had. Even the day 3 data showed a vast number of changes that we concluded that we could not decipher which changes might me related to the antiviral activity or potential 

Fig 6: Can the authors please elaborate on what kinds of effect on the viral transcriptome upon GS-5801 treatment.

Recommendation for a better representation of the host transcriptome changes observed in Fig 6(D) through highlighting specific pathways and genes on the heat map.

Pathway changes may be shown as bar graphs as downregulated and upregulated

Indeed, the language did not match the figures well. However, Fig. 6A is solely about the HBV RNA change. But to connect the viral RNA reduction to the host effects, we changed Fig 6D, to show the most up/down regulated pathways along with the change in HBV RNA. The original Fig 6D (changes in ISG transcription) is now in the supplementary material. The text now stresses the point that the vastness of the changes seen in the transcriptome upon GS-5801 treatment makes it close to impossible to infer what might be responsible for the antiviral effect. 

Hopefully, the updated manuscript tells a clear and acceptable story.

Sincerely,

Uli Schmitz, Ph.D

Executive Director, Structural Biology and Chemistry

Gilead Sciences, INC.

---

## [Editor Report · Decision Letter 1]

16 Nov 2022

Characterization of a KDM5 Small Molecule Inhibitor with Antiviral Activity against Hepatitis B Virus

PONE-D-22-15852R1

Dear Dr. Schmitz,

We’re pleased to inform you that your manuscript has been judged scientifically suitable for publication and will be formally accepted for publication once it meets all outstanding technical requirements.

Kind regards,

Haitao Guo

Academic Editor

PLOS ONE
---

## [Editor Report · Acceptance letter]

22 Nov 2022

PONE-D-22-15852R1 

Characterization of a KDM5 Small Molecule Inhibitor with Antiviral Activity against Hepatitis B Virus 

Dear Dr. Schmitz:

I'm pleased to inform you that your manuscript has been deemed suitable for publication in PLOS ONE. Congratulations! Your manuscript is now with our production department. 

Kind regards, 

on behalf of

Dr. Haitao Guo 

Academic Editor

PLOS ONE